# Reduced adult stem cell fate specification led to eye reduction in cave planarians

Luiza O. Saad[1,2,3,4], Thomas F. Cooke[2,3,4], Kutay D. Atabay[2,3,4], Peter W. Reddien ®[2,3,4] ✉ & Federico D. Brown ®[1] ✉

Eye loss occurs convergently in numerous animal phyla as an adaptation to dark environments. We investigate the cave planarian *Girardia multidiverticulata (Gm)*, a representative species of the Spiralian clade, to study mechanisms of eye loss. We found that *Gm*, which was previously described as an eyeless species, retains rudimentary and functional eyes. Eyes are maintained in homeostasis and regenerated in adult planarians by stem cells, called neoblasts, through their fate specification to eye progenitors. The reduced number of eye cells in cave planarians is associated with a decreased rate of stem cell fate specification to eye progenitors during homeostasis and regeneration. Conversely, the homeostatic formation of new cells from stem cell-derived progenitors for other tissues, including for neurons, pharynx, and epidermis, is comparable between cave and surface species. These findings reveal a mode of evolutionary trait loss, with change in rate of fate specification in adult stem cells leading to tissue size reduction.

Animals living in dark environments have evolved strikingly similar morphological traits, even among distantly related species[1,2]. Loss or regression of eyes and disruption of eye and body pigmentation are examples of morphological adaptations found in fossorial[3], faunal[4], cavernicolous[5], and abyssal[6,7] species. A key question regarding species inhabiting dark environments is whether similar molecular or cellular mechanisms are employed in various animals that have independently undergone evolutionary adaptations to such environments, including crustaceans[8], planarians[9], salamanders[10], mammals[11], and fish[12].

Current understanding of the processes underlying eye loss under distinct developmental contexts comes from only a handful of organisms amenable to developmental genetics studies, such as cavefish[12] and a few other vertebrates, crustaceans, and cave insect species[5,8,13,14]. These species belong to two of the three major bilaterian animal clades (i.e., the Deuterostomes and the Ecdysozoa). Representation of species from within the Spiralia is desired for uncovering the range of mechanisms involved in the evolution of trait loss. Planarians provide an attractive Spiralian model system for the study of the development and evolution of eyes in dark environments for

several reasons. First, many surface planarian species contain eyes with simple anatomy, being comprised of a pigmented cup epithelium and photoreceptor neurons, which send rhabdomeres into the pigment cup and axons to a bilobed brain[15,16]. Second, despite their apparent simplicity, planarian eyes use similar developmental and phototransduction-related genes to those in animals with more complex eyes, such as vertebrates or arthropods, allowing comparison between evolutionarily distant species[17–19]. Third, eyeless and depigmented planarian species have evolved multiple times in caves, allowing comparisons between multiple cave species undergoing independent evolution, and with variable times of divergence from their surface sister species[9,20,21]. Finally, planarians display the capacity to fully regenerate their eyes and to replenish eye tissue in the adult state from stem cells, presenting an intriguing setting to investigate what limits eye formation in cave species[17–19,22].

Eyes in planarians undergo continuous turnover by the constant contribution of eye progenitors derived from well-characterized stem cells called neoblasts[17,18,23]. Neoblasts, responsible for regeneration and tissue maintenance in planarians, are comprised of multiple classes that exhibit distinct gene expression signatures[24,25]. Fate specification

[1]Departamento de Zoologia, Instituto de Biociências, Universidade de São Paulo, São Paulo, Brazil. [2]Whitehead Institute for Biomedical Research, Cambridge, MA, USA. [3]Department of Biology, MIT, Cambridge, MA, USA. [4]Howard Hughes Medical Institute, Chevy Chase, MD, USA. ✉e-mail: reddien@wi.mit.edu; fdbrown@usp.br

can occur in neoblasts and is associated with the expression of distinct transcription factor signatures for each differentiated tissue type[24]. Eye-specialized neoblasts are specified in a broad prepharyngeal region and produce post-mitotic progenitors that migrate, incorporate into eyes, and differentiate[26]. This migratory targeting process is guided by adult positional information provided by position control genes[26]. Planarian eye cell differentiation requires similar sets of developmental genes across embryonic development, regeneration, and homeostasis[17–19]. *six-1/2-1*, *eya*, and *ovo* encode transcriptional regulatory proteins that specify the fate of all eye cells. The transcription factor-encoding *otxA* gene is expressed in photoreceptor neurons and their progenitors. *sp6/9* and *dlx* are expressed in the optic cup progenitors and mature optic cup cells[17–19]. Genes associated with

eye function, including *opsin* and *arrestin* for photoreceptor cells and *tyrosinase* for optic cup cells, are activated with differentiation[17,18,22].

In this work, we identified an adult mechanism that explains the reduced state of the eyes in *Girardia multidiverticulata*, involving a specific reduction of eye cell progenitor specification from stem cells. These findings show how changes in adult-specific processes of tissue maintenance and repair can occur in evolution to result in trait loss.

## Results

### *Girardia multidiverticulata* have rudimentary eyes

*Girardia multidiverticulata* is a planarian species endemic to the Buraco do Bicho cave in Mato Grosso do Sul, Brazil (Fig. 1a). According to the original description of the species, *G. multidiverticulata* is a

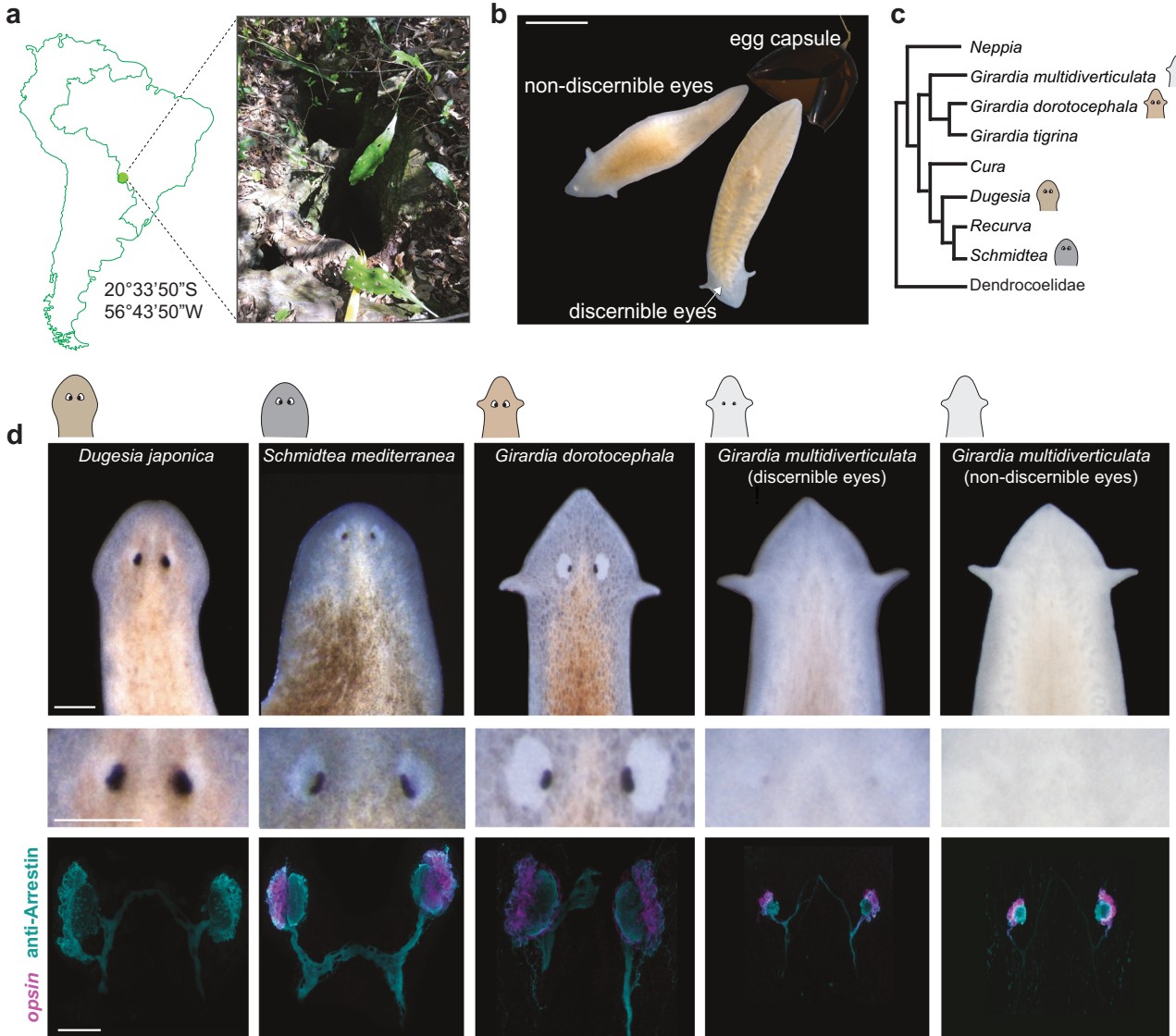

**Fig. 1 | The cave planarian *Girardia multidiverticulata* has reduced eyes and exhibits two visually distinct morphotypes. a** *G. multidiverticulata* type locality at the cave Buraco do Bicho, located in the Serra da Bodoquena, Mato Grosso do Sul, Brazil. Detail of the cave entrance. **b** Two sibling morphotypes emerging from an egg capsule; one morphotype presents small pigmented eyes (discernible eyes) and the other has no visible eyes (non-discernible eyes). Scale bar, 5 mm. **c** Phylogeny (consensus tree) of the main groups of terrestrial Dugesiidae; *G. multidiverticulata* is a sister species to other *Girardia* species, together forming a monophyletic group. The sister clade of the *Girardia* species group includes *Cura*, *Dugesia*, *Schmidtea*, and *Recurva*. Consensus tree is supported by phylogeny based

on four gene markers (COI, EF, 28S, and 18S rDNA type I and II) (Supplementary Fig. 1 and Supplementary Tables 1 and 2). **d** Comparative morphologies of the eyes in surface and cave planarians using live-imaging (top two rows), and by double labeling (bottom row) using fluorescent in situ hybridization of *opsin* mRNA in photoreceptor cells (cyan) and immunocytochemistry for Arrestin (axonal projections converge and cross the midline of the head). Similar results were observed in *G. multidiverticulata* non-discernible eyes n = 86; *G. multidiverticulata* discernible eyes n = 85; *G. dorotocephala* n = 58; *Dugesia japonica* n = 36; *Schmidtea mediterranea* n = 32. Scale bar (top two rows), 1 mm; scale bar for fluorescent images (bottom row), 50 μm.

translucent animal with no externally apparent eyes[27]. However, observations of the offspring from animals raised in the laboratory revealed the occurrence of two visually distinguishable morphotypes segregating among siblings. The 'discernible eye morphotype' displayed small, pigmented rudimentary eyes visible with light microscopy, whereas the 'non-discernible eye morphotype' exhibited no visible eyes by light microscopy (Fig. 1b).

Most cave animals are derived from surface-dwelling ancestors, allowing a direct comparison of their eye features[28]. To identify the relationship of *G. multidiverticulata* to surface-dwelling planarian species, we carried out phylogenetic analyses using concatenated mitochondrial (COI) and nuclear (18S, 28S, and EF - Elongation Factor) available sequences of Dugesiidae species (Supplementary Table 1, 2). Both Bayesian and Maximum Likelihood phylogenetic inferences placed *G. multidiverticula* inside a well-supported monophyletic *Girardia* clade[29] (Fig. 1c, Supplementary Fig. 1). *G. dorotocephala* was selected for this study as a surface control species because *(i)* it is closely related to *G. multidiverticulata*, *(ii) G. dorotocephala* is accessible and readily grown in a lab setting, and *(iii)* no other surface-dwelling *Girardia* species was available from near the cave that could represent the direct sister species to *G. multidiverticulata*. Other closely related genera (e.g., *Schmidtea* and *Dugesia*) were also used for surface species comparisons (Fig. 1c and Supplementary Fig. 1).

The presence of two large pigmented eyes is a common feature of the Dugesiidae family, as well as of the genus *Girardia* (Fig. 1d). We examined *G. multidiverticulata* with an antibody to the Arrestin protein and an RNA probe to *opsin* with fluorescence in situ hybridization (FISH), and found that both morphotypes (discernible and non-discernible eyes) have eyes (Fig. 1d), which were reduced in size compared to surface planarians (Fig. 1d). The presence of photoreceptor neurons in both cave morphotypes suggests that the non-discernible eye morphotype likely harbors a change to the pigmented optic cup.

The presence or absence of pigment in the rudimentary eyes of *G. multidiverticulata* was a consistent attribute of individuals of the two morphotypes that were maintained in separate cultures (>200 animals were observed for each morphotype over 3+ years of culturing). Animals were also crossed with individuals of the same morphotype. Pairs of non-discernible eye morphotypes and pairs of discernible eye morphotypes were isolated from hatching until reaching sexual maturity (around six months). Their offspring were isolated following multiple crosses over four generations. F3 crosses between individuals with non-discernible eyes produced 100% (46/46) of offspring with non-discernible eyes. F3 crosses between individuals with discernible eyes resulted in progeny with approximately 20% non-discernible (3/16) eyes and 80% discernible eyes (13/16). These results further suggest a genetic basis for these two distinct eye pigmentation phenotypes and the dominance of the discernible eye phenotype.

## Specificity in the reduction of photoreceptor cell number

To assess the difference in eye size between *Girardia multidiverticulata* and planarians inhabiting the surface, we counted the number of photoreceptor cells (opsin[+] and/or Arrestin[+]) in individuals of different sizes within the species *G. multidiverticulata* (discernible and non-discernible eyes) and several surface species (*Girardia dorotocephala*, *Schmidtea mediterranea*, and *Dugesia japonica*). We also sought to determine whether any reduction in cell numbers in cave planarian eyes was accompanied by a reduction in the size of other aspects of the nervous system or other body organs by measuring brain length (utilizing DAPI) and pharynx length (in live animals and in fixed animals utilizing DAPI) (Fig. 2 and Supplementary Fig. 2). In cave and surface species, the number of photoreceptor cells, brain length, and pharynx length all positively correlated with body length (Fig. 2a–c and Supplementary Fig. 2a, b). However, the slope of the simple linear regression between eye and body size in cave planarians was notably

smaller in cave planarians than in the surface-dwelling planarians (Fig. 2a). By contrast, the scaling of brain or pharynx length with animal length was similar in the cave and surface species (Fig. 2b, c and Supplementary Fig. 2b). Both cave morphotypes had significantly fewer photoreceptor cells compared to surface planarians of similar length, but displayed similar brain and pharynx sizes (Supplementary Fig. 2c–f). The two cave morphotypes showed no significant differences between each other in the number of photoreceptor neurons, or in brain or pharynx length (Fig. 2d and Supplementary Fig. 2c–f).

We also utilized the marker *pyrokinin prohormone-like 1* (*ppl-1*) to quantify a specific central nervous system cell type and found no overt difference in the number of *ppl-1*[+] cells in the brains of both *G. multidiverticulata* discernible and non-discernible eye morphotypes and *G. dorotocephala* of similar body size (Fig. 2e, f). These data indicate that the anatomical scaling changes in the cave planarians were largely specific to the eyes. These results reveal a difference between *G. multidiverticulata* and the surface-dwelling planarians in allometric scaling of the eye with respect to body length and with respect to other features such as brain size, number of *ppl-1*[+] neurons, and pharynx length.

## Cave planarian eyes are functional

Light perception can vary in distinct cave animal species. Several blind cavefish show no preference for light or darkness[30], whereas some other cave species still retain the ability to detect light, such as troglobiont crayfish, beetles, and opiliones[31–34]. Surface planarians are well known to display negative phototaxis[35,36]. To assess light intensity perception in cave planarians we performed behavioral experiments using a light gradient assay[26]. Planarians were placed in an arena with a gradient of different light intensities, and behavior was compared to that in a uniformly gray control arena (Supplementary Fig. 3a, b). Cave planarians displayed negative phototaxis, similar to the surface planarian *G. dorotocephala*, showing a preference for darker regions in the light intensity gradient (Fig. 3a). Both species showed no positional preference in the arenas with uniform light in control experiments. In addition, no main differences in behavioral responses to light were detected between the two cave morphotypes (Fig. 3a).

Studies on the blind cavefish *Astyanax mexicanus* showed an absence of light detection from their eyes, but some ability to detect light associated with the presence of the pineal organ[37]. It has also been suggested that planarians display some extraocular response to different wavelengths of light[38,39]. To investigate if the negative phototactic response of cave planarians is dependent on the presence of eyes we resected the eyes of cave and surface planarians (Fig. 3b, c). Eye-resected animals showed no preference for darker regions in the light intensity gradient arena, indicating that cave planarian light avoidance involves an ocular response. The slight preference for darker regions observed in eye-resected *G. multidiverticulata* individuals from the non-discernible eye morphotype was attributed to incomplete eye resection, associated with the challenge of resecting eyes that were not readily visible by light microscopy; Arrestin[+] cells were present one day after eye removal, confirming incomplete resection (Fig. 3c). Nevertheless, the results with the discernible eye morphotype indicate that the *G. multidiverticulata* light response in this assay was eye dependent (Fig. 3b).

Surface-dwelling planarians exhibit a range of phototactic behaviors in response to various wavelengths of light[40]. Shorter wavelengths (e.g., blue and green) trigger pronounced avoidance behavior, whereas longer wavelengths (e.g., red) do not[40]. Given that cave-dwelling planarians possess smaller eyes, we investigated whether they could discern light intensity gradients across different wavelengths using optical filters. These filters isolate specific regions of the light spectrum while transmitting only the wavelength of interest[41]. As expected, the surface-dwelling planarian, *G. dorotocephala*, displayed a robust photophobic response to green and blue light but exhibited no reaction to red light (Fig. 3d, Supplementary Fig. 3c). By contrast,

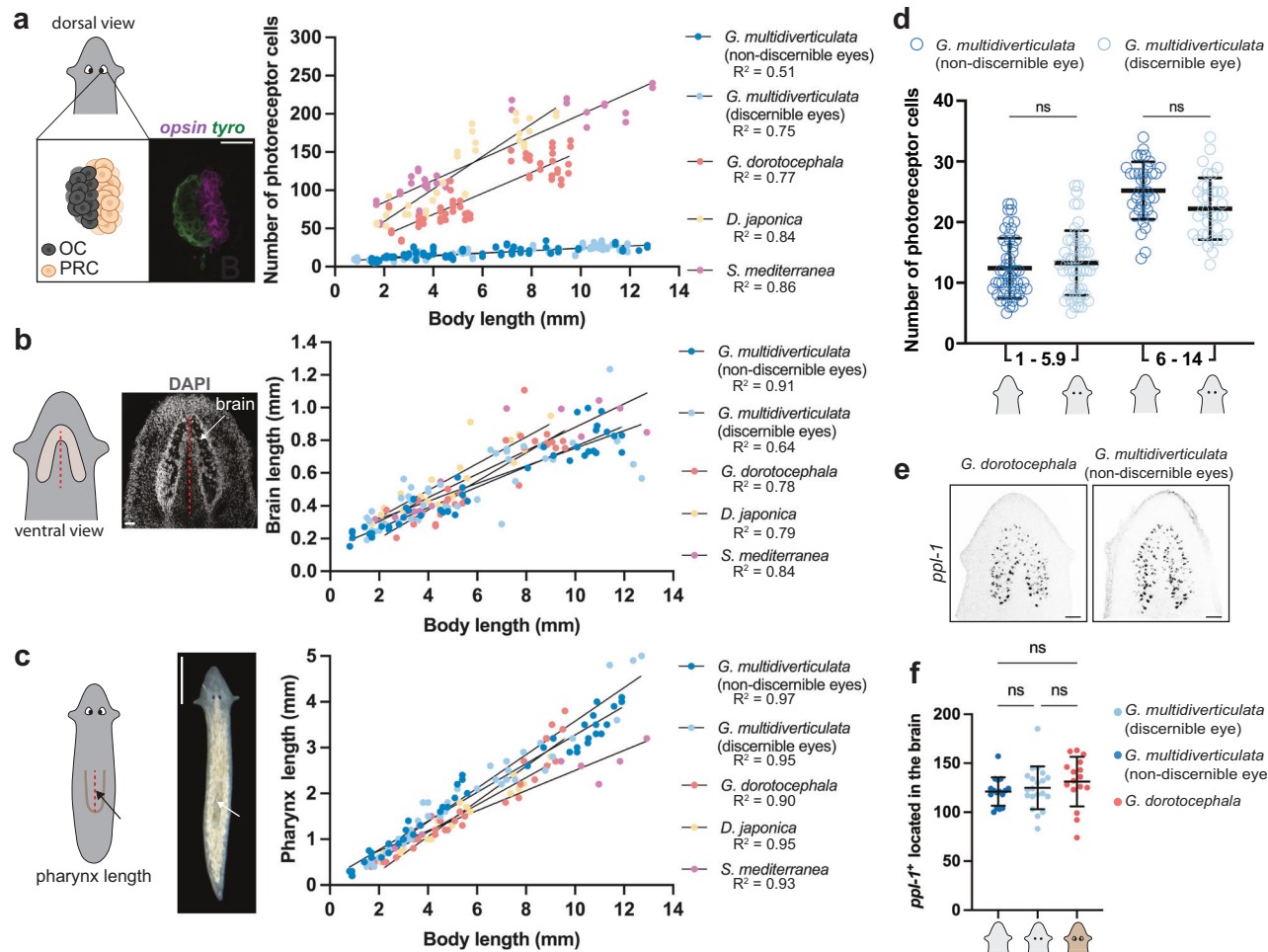

**Fig. 2 | Eye-body allometry: the cave planarian *Girardia multidiverticulata* displays fewer photoreceptor cells than surface planarians, but similar brain and pharynx size. a** Schematic representation of eye photoreceptor cells with details of how photoreceptor cells were identified and counted. Simple linear regressions demonstrating cave planarian morphotypes present small numbers of photoreceptor cells that increase more slowly with body size when compared to the steeper slope observed for all three species of surface planarians (*Gm*, *G. multidiverticulata* non-discernible eye *n* = 86; *Gm(d)*, *G. multidiverticulata* discernible eyes *n* = 85; *Gd*, *G. dorotocephala n* = 58; *Dj*, *Dugesia japonica n* = 36; *Sm*, *Schmidtea mediterranea n* = 32). Scale bar, 50 μm. **b** Schematic representation of brain anatomy and the landmarks used for brain length measurements. All planarians (cave and surface species) display similar brain length-body size allometric scaling (*Gm* non-discernible eyes *n* = 43; *Gm* discernible eyes *n* = 43; *Gd n* = 29; *Dj*

*n* = 18; *Sm n* = 16). **c** Schematic representation of pharynx length live measurements. Simple linear regressions demonstrating cave and surface planarians have similar pharynx length-body size allometric scaling (*Gm n* = 43; *Gm*(d) *n* = 43; *Gd n* = 29; *Dj n* = 18; *Sm n* = 16). Scale bar, 1 mm. **d** Both cave morphotypes have a similar number of photoreceptor cells in animals with small (1–5.9 mm in length) and large (6–14 mm in length) body sizes. *G. multidiverticulata* morphotypes present no differences in eye-body allometric scaling between each other (*Gm* non-discernible eyes *n* = 86; *Gm* discernible eyes *n* = 85). **e** *ppl-1*+ cells in the brains of *G. dorotocephala* and *G. multidiverticulata*. **f** Similar overall number of *ppl-1*+ cells in the brains of *G. dorotocephala* (*n* = 16) and *G. multidiverticulata* discernible (*n* = 16) and non-discernible eye (*n* = 20) morphotypes. Scale bar, 50 μm. For **d** and **f** Mean ± SD and one-way ANOVA followed by Tukey's multiple comparisons test. Mean ± SD. R, coefficient of determination. ns not significant p > 0.01.

both cave planarian morphotypes lacked a strong light response across all tested wavelengths, except for a few responses to green (3/10) and blue (3/10) (Fig. 3d). Although these animals exhibited a tendency to be located in the darker areas in the blue light assay, they appeared unable to effectively resolve the light gradient (Fig. 3d). These findings suggest that *G. multidiverticulata* lack a response to red wavelengths of light, and do not exhibit a pronounced photophobic response to green or blue in the assay utilized. The reduced eye size in these cave-dwelling animals is likely related to its reduced light sensing behavior, allowing them to moderately detect stimuli in the full visible light spectrum, but to struggle to respond to particular light wavelengths.

**Conservation of eye formation programs in *G. multidiverticulata***
Given the well-documented expression defects of core eye developmental genes (e.g., *otx*, *pax6*, and *crx*) in various cave-dwelling taxa (e.g., fish, salamander, and crustaceans[12,42–48]) it was of interest to

determine if the eye formation program in cave planarians also exhibited any gene expression differences that might explain the small eye size. Fluorescent in situ hybridization (FISH) and in situ hybridization chain reaction (HCR) were used to investigate the expression of genes related to planarian eye development. Similar to the case of surface planarians, the two eye cell types in both cave planarian morphotypes were confirmed to express *opsin* and *arrestin* in photoreceptor cells and *tyrosinase* in optic cup cells (Fig. 4a, Supplementary Fig. 4a). *tyrosinase* is normally expressed in optic cup cells for melanin production and was expressed in the eyes of both cave morphotypes. Additionally, cave planarians expressed the planarian eye transcription factor-encoding genes *ovo*, *six-1/2-1*, and *eya* in photoreceptor neurons and optic cup cells, and *sp6/9* in optic cup cells (Fig. 4a). The photoreceptor cells also expressed *otxA* and *klf*, as is the case for *S. mediterranea* (Fig. 4a). Unexpectedly, *dlx* in cave planarians was expressed only in the photoreceptors (Fig. 4a) and expression was not detected in the optic cup cells, unlike what was previously described for surface

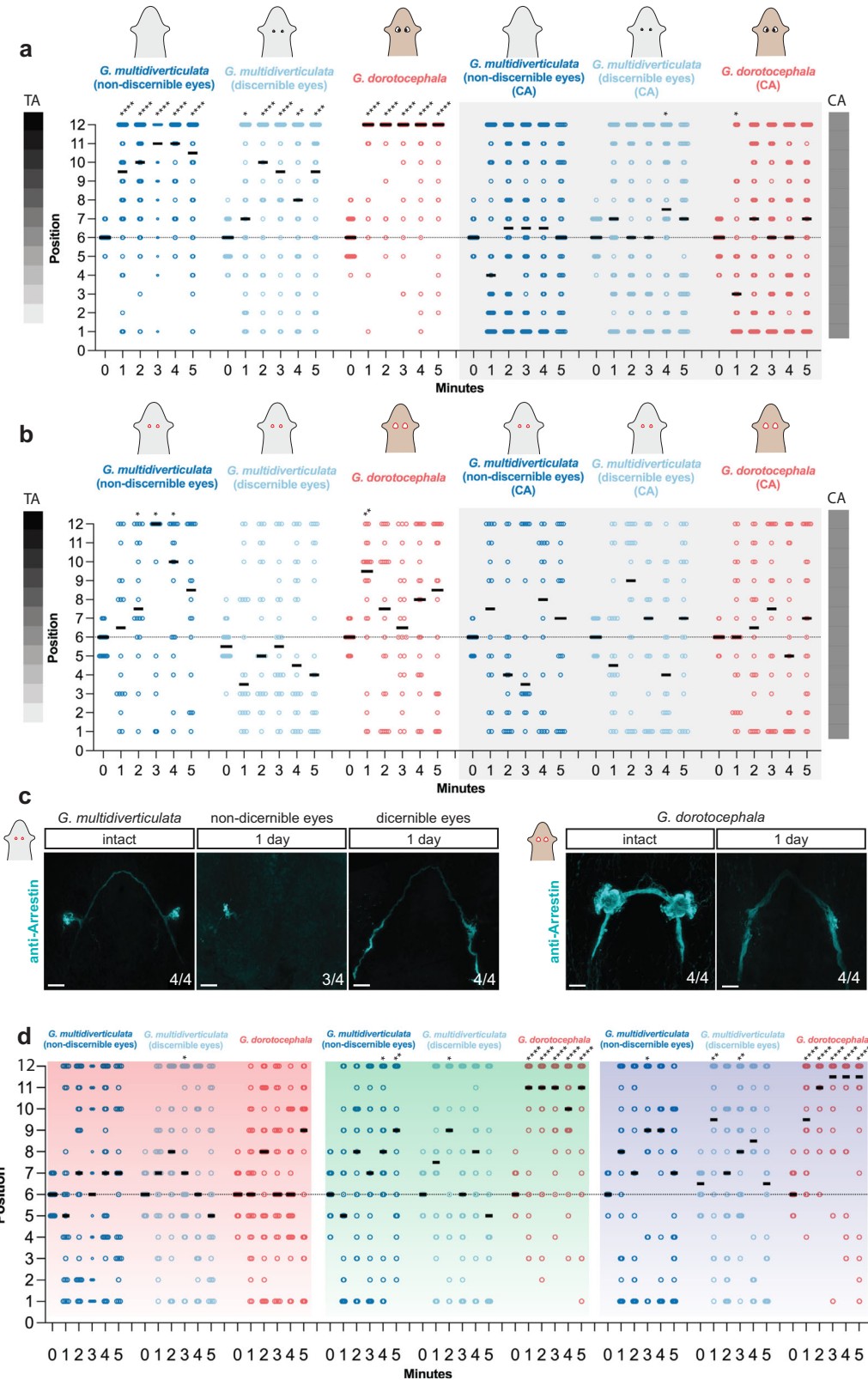

planarians[17,18]. Another difference was in the expression of *foxQ2*, which was expressed in anterior photoreceptor cells as well as in other unidentified anterior cells (Fig. 4a). Previous work in surface planarians has only shown the expression of *foxQ2* co-localized with photo-receptor cells and not in any other anterior cell near the eyes[18].

Previous studies in surface planarians demonstrated that inhibition of *ovo, six-1/2-1, eya, dlx, otxA*, or *foxQ2* genes also resulted in eye

malformations, leading to a reduced number of photoreceptor cells or eye loss[17,18]. RNA interference (RNAi) experiments showed that *ovo, six-1/2-1*, and *eya* are necessary for eye formation in *G. multidiverticulata*, because downregulation of these genes resulted in animals without eyes or with malformed eyes (Fig. 4b). More specifically, *eya* inhibition resulted in the complete absence of eyes in all treated animals (n = 10). Inhibition of the other central regulators of eye formation (*ovo, six-1/2-1*)

**Fig. 3 | Eyes of the cave planarian *Girardia multidiverticulata* can mediate negative phototaxis. a** Cave *G. multidiverticulata* and surface *G. dorotocephala* animals present negative phototaxis, moving to darker regions in the test arena (three sets of experiments on the left), or no phototaxis by distributing randomly in the control arena (three sets of experiments on the right) (*n* = 50 animals of each species per set). **b** Eye-resected animals show no phototaxis and remain randomly distributed in both arenas (*n* = 22 animals of each species per set, except for the *G. multidiverticulata* non-discernible eye morphotype where *n* = 18); negative phototaxis in *G. multidiverticulata* occurred in some cases (*), in which eyes had not been completely removed. **c** Arrestin antibody labeling, marking the photoreceptor cells, is still present on day one after eye resection in the *G. multidiverticulata* non-discernible eyes morphotype and photoreceptor cells are absent after resection in the *G. multidiverticulata* discernible eye morphotype and in *G. dorotocephala*. Scale

bar, 50 μm. **d** Overall photophobic responses for red, green, and blue light. The negative phototaxis response was indicated by increased localization in position number 12, the darkest position in the light gradient arena. *G. dorotocephala* displayed a random distribution across the arena in red but exhibited strong negative phototaxis in green and blue arenas. *G. multidiverticulata* non-discernible eye and discernible eye morphotypes displayed little significant negative phototaxis in red, three significant responses in green, and three significant responses in blue (red filter: *Gm* non-discernible eyes *n* = 25; *Gm* discernible eyes *n* = 25; *Gd n* = 27, green filter: *Gm* non-discernible eyes *n* = 25; *Gm* discernible eyes *n* = 26; *Gd n* = 24; blue filter: *Gm* non-discernible eyes *n* = 25; *Gm* discernible eyes *n* = 26; *Gd n* = 26, animals per set). For **a**, **b**, and **d** statistical significance: one-side t-test comparing each column mean with a hypothetical value of 6 corresponding to chance. CA control arena; TA test arena, *p < 0.05, **p < 0.01, ***p < 0.001, ****p < 0.0001.

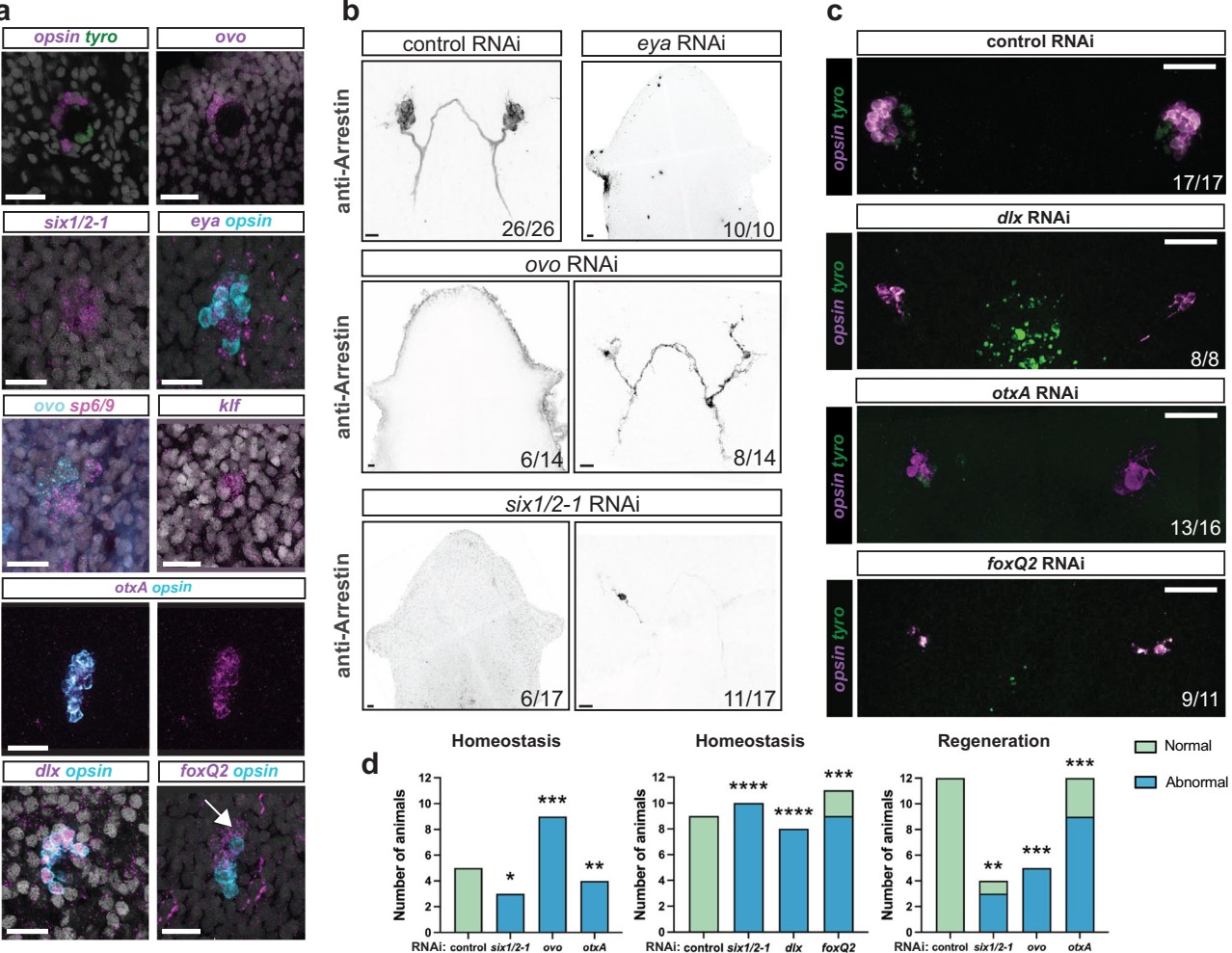

**Fig. 4 | The developmental program of planarian eye differentiation is largely conserved in cave planarians. a** Expression of key eye transcription factors in the intact cave planarian *G. multidiverticulata*, non-discernible eye morphotype. *dlx*[+] and *opsin*[+] double-positive signal shows colocalization in photoreceptor cells and not in the optic cup. *foxQ2* and *opsin* show colocalization, but *foxQ2* expression also occurs in a more anterior region near the photoreceptor cells (arrow). Representative images from *n* = 5 animals. **b** RNAi experiment phenotypes presenting normal (control), abnormal (*ovo*, *six1/2-1* right panels), or absent (*eya*, *ovo*, *six1/2-1* left panels) eyes. In control RNAi animals, 26 out of 26 presented normal eyes (26/26). Normal eyes contained photoreceptor neurons and photoreceptor axons visualized with the anti-Arrestin (VC-1) antibody. Six out of 14 *ovo* RNAi animals

displayed no eye formation (6/14), and eight out of 14 displayed an abnormal eye (8/14), containing few photoreceptor cells and disorganized axon projections. Similarly, some treated animals with *six1/2-1* RNAi displayed no eye formation (6/17), but the majority displayed abnormal eyes (11/17). **c** Abnormal phenotypes after eye transcription factor RNAi. Eye images shown of each experimental condition: Control (*n* = 26), *six1/2-1* (*n* = 17), *dlx* (*n* = 8), *ovo* (*n* = 22), *otxA* (*n* = 16), and *foxQ2* (*n* = 11). **d** RNAi during regeneration and homeostasis were blindly scored by two independent scorers, who classified the images as either normal (green) or abnormal (blue). The results were then compared using a Two-sided Fisher's exact test, **p < 0.01, ***p < 0.001, ****p < 0.0001. RNAi experiments were performed with the *G. multidiverticulata* non-discernible eye morphotype.

resulted in a slightly different outcome. Some RNAi animals exhibited a complete absence of eyes (Fig. 4b, *ovo* and *six-1/2-1*, left panels) or the presence of a small number of photoreceptor cells (-one or two Arrestin⁺ cells), with severe anatomical malformation (Fig. 4b, *ovo* and *six-1/2-1*, right panels). By contrast, inhibition of the genes encoding other eye-associated transcription factor-encoding genes, *dlx*, *otxA*, and *foxQ2*, did not block eye formation entirely, but instead resulted in eyes that were frequently malformed (Fig. 4c). Unexpectedly, RNAi of the *dlx* gene affected not only the development of photoreceptor cells, but also the optic cup cells (Fig. 4c). Additionally, RNAi of the *foxQ2* gene, which was previously reported to impact only the number of photoreceptor cells[18], was found to influence overall eye formation in cave-dwelling planarians, resulting in abnormal eyes presenting few photoreceptor cells, asymmetric eyes, or misshapen eyes. Eyes in the control, as well as in the *six1/2-1*, *dlx*, *ovo*, *otxA*, and *foxQ2* RNAi treated animals were classified as either normal or abnormal by two independent scorers, who were blinded to the experimental condition (Fig. 4d). Overall, these findings indicate that the expression and function of eye-related transcription factors in eye regeneration are largely conserved between surface species and *G. multidiverticulata*, except for more general eye development defects observed in *dlx*, *otxA*, and *foxQ2* RNAi animals. These cases show that development of optic cup and/or photoreceptor cells in the eyes of *G. multidiverticulata* may be more sensitive to specific molecular perturbations than *S. mediterranea*.

The downregulation of two highly conserved transcription factors of animal eye development (*Pax6A* and *Rx3*) has been shown to be involved in the development of reduced eyes in cavefish[49–52]. However, *Pax6A* and *Rx*-related genes were shown to not have an involvement in surface planarian eye formation[18,19,53,54]. We inhibited orthologs of both of these genes by RNAi in *G. multidiverticulata* nonetheless, and eye formation was not disrupted (Supplementary Fig. 4b). As is the case in *S. mediterranea, Pax6A* was expressed only in brain cells in *G. multidiverticulata* (Supplementary Fig. 4c). These results support the prior conclusion that these genes are dispensable for planarian eye development.

To further investigate potential gene expression differences between cave-dwelling and surface-dwelling planarians, we performed RNA sequencing with individual isolated eyes from multiple species, including both morphotypes of *G. multidiverticulata*, *G. dorotocephala*, *S. mediterranea*, and *D. japonica*. We then conducted differential gene expression analysis between the species for predicted orthologs of 140 genes previously identified as expressed in the eyes of *S. mediterranea*[18]. These genes included those predicted to be associated with eye function (*e.g.*, phototransduction, cGMP pathway activation, signaling receptors, solute transport mechanisms, kinases, and melanin synthesis)[18] (experimental pipeline described in Supplementary Fig. 4d).

The average expression of a conserved set of eye-related transcription factors (i.e., *ovo, six 1/2-1, eye, otxA, sp6/9, dlx, soxB1-1, meis, foxQ2*, and *klf*), as well as other previously described eye-related genes, indicated that no major differences existed between cave and surface-dwelling species (Fig. 5a and Supplementary Fig. 5; with individual eyes sequenced, similar expression levels indicate similar levels per cell of each eye). Differential gene expression results corroborated these findings (Supplementary Table 3). Eight genes associated with eye function exhibited altered expression levels in cave planarian species (including both morphotypes of *G. multidiverticulata*) when compared to their surface-dwelling counterparts (*19866* and *crf-r* were upregulated and *22592, arrestin, cng, cpo, rops2*, and *tph* were downregulated; padj <0.05 and log2FoldChange > 1) (Fig. 5b, Supplementary Table 3). *19866, cng, cpo, arrestin*, and *rops2* are known to be expressed in planarian photoreceptor cells, and *crf-r, 22592*, and *tph* are expressed in planarian optic cup cells[18]. *19866* encodes the planarian ortholog for *thada –thyroid adenoma associated*, and *cpo* encodes an ortholog of the Couch potato protein[18]. Mutation of both genes in *Drosophila*

results in nervous system defects[55,56]. *cng* encodes a Cyclic nucleotide-gated (CNG) channel, which mediates membrane depolarization. CNG channels were initially identified in retinal photoreceptors and olfactory sensory neurons and have been hypothesized to regulate light response or adaptations to light/dark conditions in mouse and human[57–60]. Arrestin proteins inhibit signal transmission after photoreceptor light activation[61]. Similarly, *rops2* belongs to the rhodopsin family and encodes peropsin, which is a retinal pigment epithelium-derived rhodopsin found in various animals[62]. In planarians, peropsin is expressed not only in the eyes but also in other cells in the body, potentially playing a role in extraocular light perception[18,38]. *crf-r* encodes a protein similar to a calcitonin gene-related peptide type 1G protein-coupled receptor (GPCR)[63]; *22592* also encodes a predicted GPCR protein[18]. Finally, tryptophan hydroxylase (*tph*) encodes a serotonin biosynthesis enzyme[64] involved in planarian eye pigmentation[65]. Overall, cave planarians differ from surface planarians in the expression levels of eye-related genes involved in light transduction, regulation related to transmembrane GPCRs, and pigment formation. These findings indicate that changes in the expression of eye-related genes occurred during the divergence of cave and surface planarian species, along with a reduction in the number of eye cells observed in cave planarians.

Next, we compared gene expression between the two cave morphotypes. Both displayed very similar overall gene expression patterns, and principal component analysis (PCA) grouped the samples together in a single cluster (Supplementary Fig. 4e). Pair-wise differential gene expression analyses identified ten genes that were differentially expressed: *8711* and *qdr* genes were upregulated in the non-discernible eye morphotype, and *11992, 16656, 25626, 29449, 59sley, cgs-pde, soxB1-1*, and *tyro* genes were downregulated (padj <0.05 and log2FoldChange > 1) (Fig. 5c, Supplementary Table 3). *8711, 11992, 16656, 25626, 29449*, and *cgs-pde* are all known to be expressed in planarian photoreceptor cells. Planarian pigment cup cells express *qdr* and *tyro*. *59sley* is expressed in both the pigment cup and photoreceptor cells of *S. mediterranea*[18]. Altogether, orthologs of these genes have roles in phototransduction, as well as in light perception or behavioral responses[18,66–69]. *soxB1-1* was downregulated in the non-discernible eye morphotype. *soxB1-1* is expressed in *S. mediterranea* anterior photoreceptors contributes to promoting the differentiation of specific subsets of photoreceptor neurons during eye regeneration[18]. The *tyro* gene encodes *tyrosinase*, which plays a vital role in melanin biosynthesis[70]. Although no significant difference was found in the expression of *tyrosinase* between the two morphotypes with FISH, differences in *tyrosinase* transcript levels were observed in the sequencing data and could potentially contribute to less distinct eye pigmentation levels (Supplementary Fig. 4a).

## A reduced rate of new eye cell formation in adult cave planarians

Given the conservation of the eye formation program in cave planarians and the absence of a striking difference in eye-associated gene expression, we sought to identify underlying factors contributing to the eye reduction observed in *G. multidiverticulata*. Adult tissues and organs in planarians are maintained by cellular turnover. The number of newly differentiating cells and dying cells will determine steady-state eye size[22,71]. We, therefore, investigated these dynamics in *G. multidiverticulata*. Neoblasts are the only dividing somatic cells in adult planarians[72]. EdU can therefore be specifically incorporated into neoblasts and then remain present in the labeled progeny of neoblasts that differentiate. Animals of similar size were immersed in F-ara-EdU (EdU) for 48 h, and then fixed at 4, 8, and 12 days to quantify cells that were co-labeled with EdU and the different cell markers to calculate the rates of new cell incorporation. Cave and surface planarians exhibited comparable numbers of overall EdU-positive cells across the time points, indicating a similar overall rate of progenitor production in the body (Fig. 6a, Supplementary Fig. 6a, b). By contrast, cave planarians

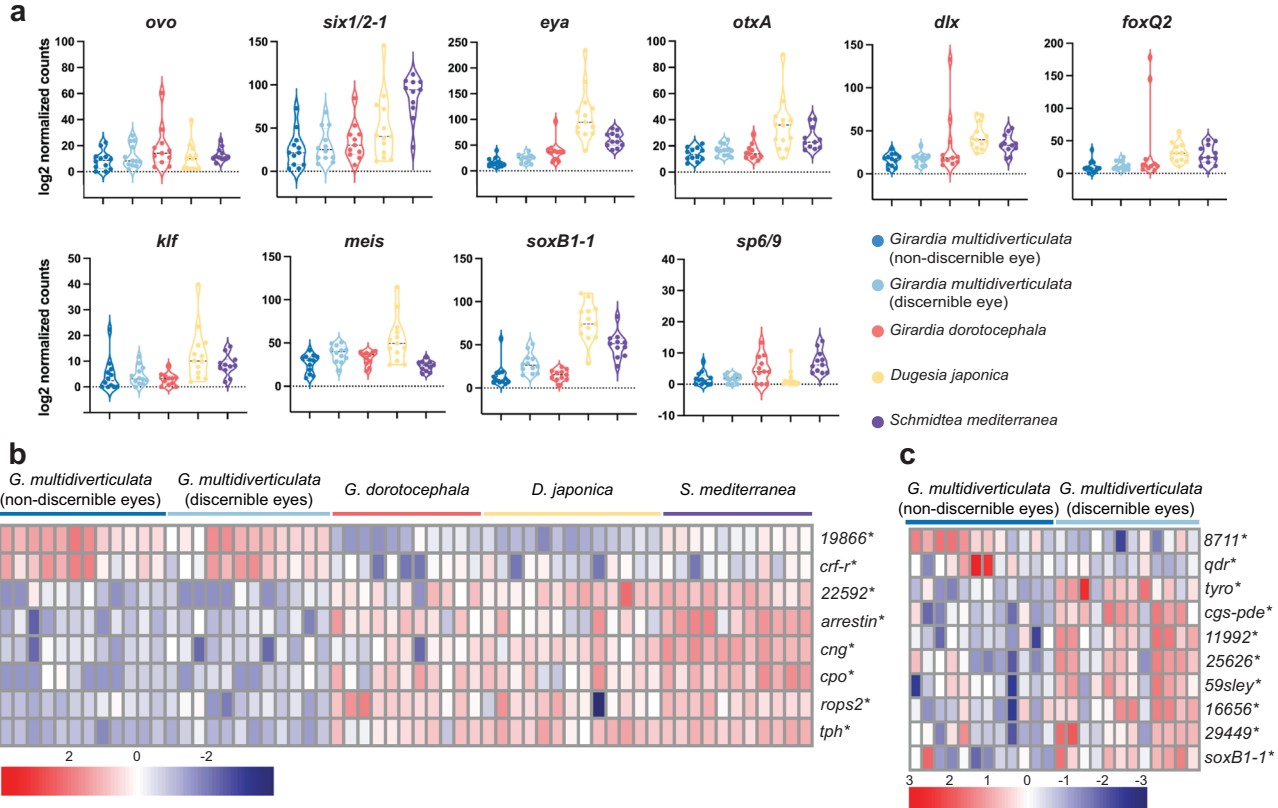

**Fig. 5 | Differential gene expression analysis of eye-related genes in cave and surface planarians. a** Violin plots of DESeq2 log$_2$ normalized counts of eye-related transcription factors that did not display significant differences. Each dot represents the log normalized counts from a single eye ($Gm$ $n$ = 12; $Gd$ $n$ = 11; $Dj$ $n$ = 13; $Sm$ $n$ = 11). **b** Heatmap of selected eye-related genes that were significantly different in expression between cave and surface planarians. Both the *G. multidiverticulata* non-discernible eye morphotype and the *G. multidiverticulata* discernible eye

morphotype served as the reference for comparison. *padj <0.05. DESeq2 Wald test pair-wise. ($Gm$ $n$ = 12; $Gd$ $n$ = 11; $Dj$ $n$ = 13; $Sm$ $n$ = 11). **c** Heatmap of differentially expressed eye-related genes from DESeq2; Wald test pair-wise comparison between the two cave morphotypes, ($n$ = 12) *padj <0.05. Heatmap shows upregulated (log2FoldChange > 1) or downregulated (log2FoldChange < −1) genes; (red) up, (blue) down. Scale bars, 50 μm. Complete statistical results presented in Supplementary Table 3.

incorporated substantially fewer new cells into the eyes (EdU$^+$; *arrestin*$^+$ cells) when compared to surface planarians (Fig. 6b, d). Because the two cave planarian morphotypes (discernible and non-discernible eyes) showed no significant differences in the incorporation of new eye cells, we combined the results of both morphotypes for the *G. dorotocephala* comparisons (Supplementary Fig. 6c).

We also quantified progenitor cell incorporation into brain cell populations using EdU and *ppl-1* (EdU$^+$; *ppl-1*$^+$ cells) (Fig. 6c, e), as well as EdU and the ciliated neuronal marker *pkd1l-2* (EdU$^+$; *pkd1l-2*$^+$ cells) (Fig. 6f, g and Supplementary Fig. 6d). In contrast to the eyes, both cave and surface species had a similar amount of new brain cells and ciliated neuronal progenitor incorporation (Fig. 6e, g).

**A lower rate of eye fate choice in adult stem cells in cave planarians**

The reduced rate of eye cell production in cave planarians could in principle be explained by a reduction in the probability of stem cell fate specification toward eyes. To investigate this possibility further, we first quantified whether *G. dorotocephala* and *G. multidiverticulata* exhibit similar neoblast numbers during homeostasis, using 3-dimensional reconstructed confocal whole-mount FISH z-stacks and Imaris software. In both the cave and surface species, *piwi-1*$^+$ neoblasts represented approximately 25% of all DAPI$^+$ cells in the region counted, indicating similar relative abundances of neoblasts in both species (Fig. 7a, Supplementary Fig. 6e, f, Supplementary Movies 1 and 2).

We next utilized HCR FISH and probes to *ovo* to quantify eye progenitors outside of the eye. Eye progenitors initiate in the

prepharyngeal region of *S. mediterranea* with the expression of *ovo* and other eye-associated transcription factors in scattered neoblasts[18,22]. As anticipated, there was a significantly lower number of eye progenitor cells observed in *G. multidiverticulata* compared to *G. dorotocephala* (Fig. 7b, c). To determine whether the decreased number of eye progenitors was specific to the eye stem cell fate, or could alternatively reflect a lower overall rate of progenitor production, we assessed pharynx progenitor production using a probe to *FoxA*. *FoxA* is expressed in a subset of pharyngeal neoblasts and also in some differentiated cells, prominently in the pharynx itself[73–75] (Fig. 7d, Supplementary Fig. 7a). The number of *FoxA*$^+$ presumptive pharynx progenitor cells counted in a region just anterior to the pharynx but excluding the pharynx itself was similar between *G. multidiverticulata* and *G. dorotocephala* (Fig. 7d, e and Supplementary Fig. 7a). *FoxA*$^+$ cells in this region are known to prominently include pharynx progenitors, but other *FoxA*$^+$ cells could possibly be present. Neoblasts can be depleted largely specifically by irradiation (Supplementary Fig. 7b)[76] and *G. multidiverticulata* animals four days post-irradiation exhibited a strong reduction in *FoxA*$^+$ cells in this region, consistent with an interpretation that counted cells prominently included presumptive pharynx progenitors (Supplementary Fig. 7a). Next, we compared the number of epidermal progenitors between the two species, marked by the late epidermal progenitor-specific marker glycine amidinotransferase *agat-3*[77,78]. Cells that expressed *agat-3*$^+$ were found close to the ventral and dorsal epidermis of *G. multidiverticulata* and *G. dorotocephala*, and *agat-3* signal was also detected in the *G. multidiverticulata* intestine (Fig. 7f, g, Supplementary Fig. 6g). *G.*

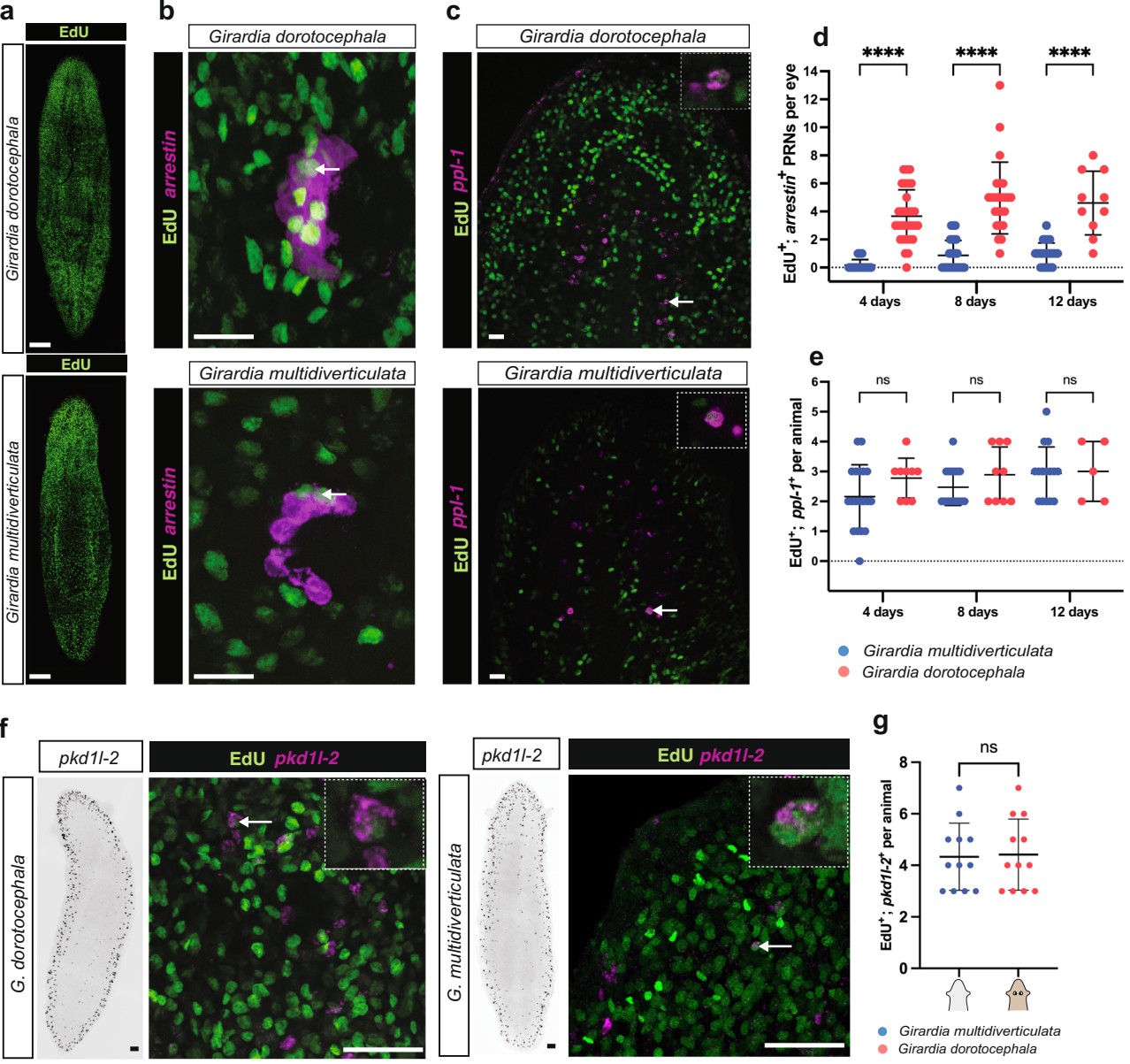

**Fig. 6 | Reduced eye cell incorporation in cave planarians. a** Similar distribution of EdU incorporation 4 days after delivery in surface and cave planarians (see also Supplemental Fig. 6a, b for additional time points and quantification). Representative images from *n* = 5 animals for each species. **b** Newly differentiated eye cells from stem cells in *G. dorotocephala* and *G. multidiverticulata* were evaluated using *arrestin* fluorescent in situ hybridization (FISH) and EdU double labeling. Arrows indicate double-positive EdU⁺; *arrestin*⁺ photoreceptor neurons (PRNs). Representative images from *n* = 25 animals for each species. **c** Newly differentiated brain cells from stem cells in surface and cave planarians were evaluated using *ppl-1*⁺ and EdU⁺ immunofluorescence double-labeling. Arrows indicate EdU⁺; *ppl-1*⁺ double positives in one visual plane with magnified examples in the right upper corner. Representative images from *n* = 20 animals for each species. **d** During homeostatic replacement of eye cells, PRN incorporation is lower in cave animals (*n* = 46, 44, 28, from 4, 8, and 12 days respectively) than in surface animals (*n* = 30,

24, 10, from 4, 8 and 12 days respectively). Intervals were compared with a Student's two-tailed t-test. **e** During homeostasis, new brain cell incorporation was similar in the two species (*Gm n* = 19, 19, 15, for 4, 8, and 12 days respectively; *Gd, n* = 9, 9, 5, for 4, 8 and 12 days respectively). Intervals were compared with a Student's two-tailed t-test. **f** Expression of *pkd1l-2* in whole animals (left panels) and one optical plane of newly differentiated ciliated neuronal cells in surface and cave planarians were evaluated using *pkd1l-2*⁺ and EdU⁺ double labeling (right panels). Arrows indicate EdU⁺; *pkd1l-2*⁺ double positives with magnified examples in the right upper corners. Representative images from *n* = 12 animals for each species. **g** During homeostasis, new ciliated neuronal cell production rate was similar in the two species (*Gm n* = 12, *Gd n* = 12, immersed in F-ara-EdU for 48 h and then fixed at 4 days). Intervals were compared with a Student's two-tailed t-test. For all graphs ****p < 0.0001; ns, not significant. Error bars represent mean ± SD. Scale bars, 50 μm.

*multidiverticulata* and *G. dorotocephala* possessed similar numbers of these epidermal progenitor cells (Fig. 7f, g). These findings are consistent with a model in which there is a specific reduction in stem cell-based progenitor production for eyes during the evolution of *G. multidiverticulata*.

We next assessed whether the reduced eye size in cave planarians could also involve modification to the rate of cell loss from eyes (eye cell death). Tissues of irradiated animals will continue to undergo

naturally occurring cell death and animals will ultimately die[76,79]. To evaluate eye cell loss by naturally occurring cell death, we quantified the number of photoreceptor cells in the eyes of animals at day 0, 8, and 12 days after irradiation of size-matched animals to assess the cell-loss rate (Fig. 7h–j; Supplementary Fig. 7b, c). Cave and surface planarians both displayed a sustained reduction in eye cell numbers over time (Fig. 7h–j). Absolute numbers of eye cells lost per eye over time were greater in surface planarians (Fig. 7i), associated with the fact that

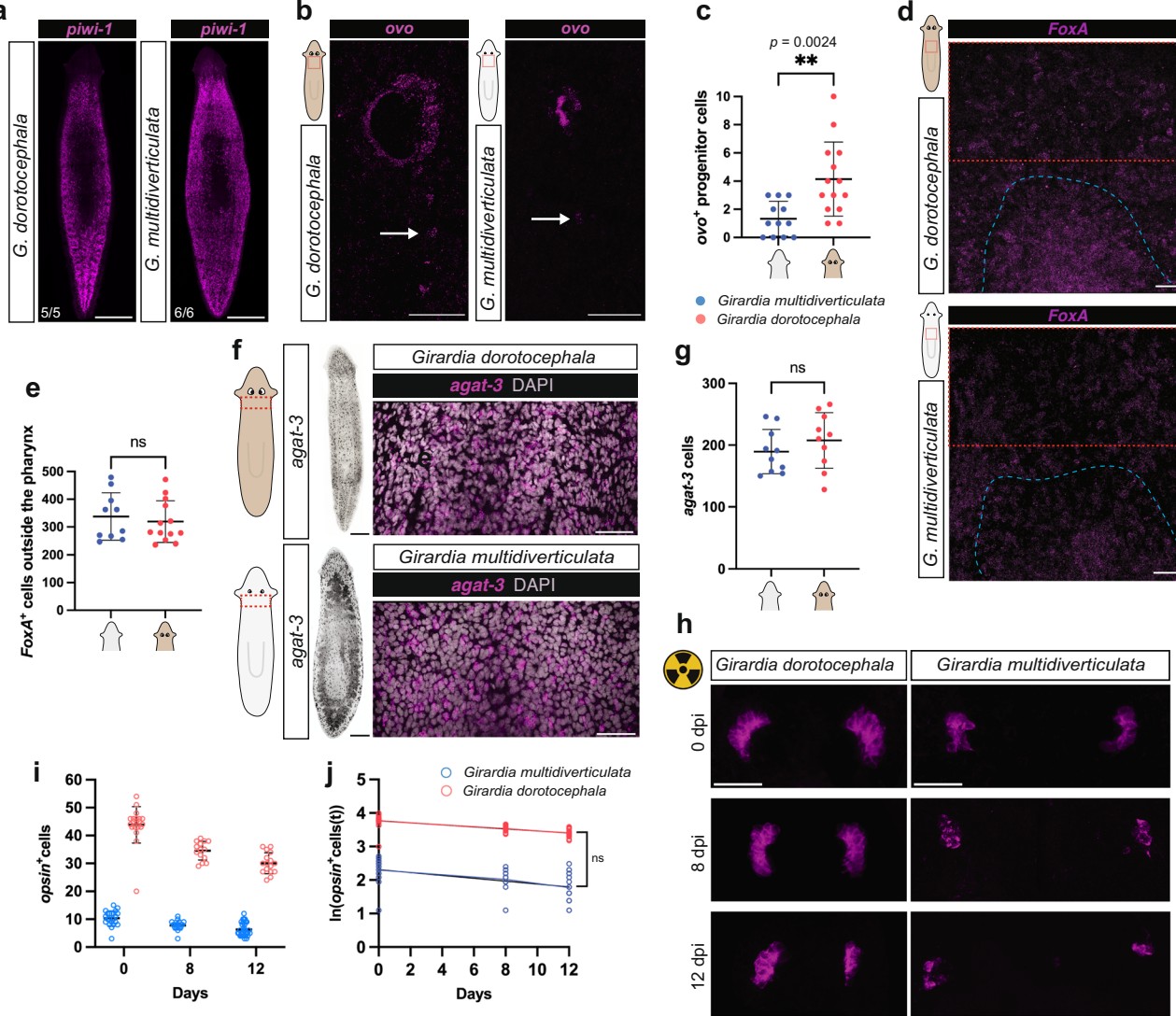

**Fig. 7 | Reduced eye progenitor cells in cave planarians. a** Whole animal *piwi-1*[+] images show similar distributions of neoblasts between *G. dorotocephala* and *G. multidiverticulata* (see also Supplementary Fig. S6e, f for quantification). **b** *G. multidiverticulata* has less *ovo* progenitor cells than *G. dorotocephala*. **c** During homeostasis *G. multiverticulata* (*n* = 12) has a significantly lower number of eye progenitor cells when compared with *G. dorotocephala* (*n* = 14). **d** *FoxA* is expressed in the pharynx (dashed blue) and in pharynx progenitors. Red dashed boxes indicate counting region. **e** During homeostasis *G. multiverticulata* (*n* = 10) has a similar number of *FoxA*[+] progenitors when compared with *G. dorotocephala* (*n* = 13). **f** FISH expression of *agat-3* in late-stage epidermal progenitor cells close to the ventral and dorsal epidermis of *G. multidiverticulata* and *G. dorotocephala*, and in the intestine of *G. multidiverticulata*. **g** During homeostasis *G. multiverticulata* (*n* = 10) has a similar number of *agat-3*[+] when compared with the *G. dorotocephala* (*n* = 10). **h** PRN (*opsin*[+] cells) loss during 12 days after irradiation. **i** PRN loss quantification corresponding to cells in (**j**) is shown. **j** After normalization, simple linear regression of log-transformed PRN (*opsin*[+]) numbers show a similar decay (slopes) after irradiation in both cave (*n* = 22, 18, 31, from 0, 8, and 12 days respectively) and surface planarians (*n* = 20, 14, 16, from 0, 8 and 12 days respectively) (p = 0.2170). Abbreviations: PRNs, photoreceptor neurons. Scale bars, 50 μm. Mean ± SD. For **c**, **e**, **g**, **i**, **j** intervals were compared with a Student's two-tailed t-test., **p < 0.01 ns, not significant. *G. multidiverticulata* morphotypes were combined in all the experiments unless otherwise specified. Size-matched animals were used in all experiments.

surface animals have a greater number of eye cells (larger eyes) available to undergo cell death. However, the loss rates per eye cell between *G. multidiverticulata* and *G. dorotocephala* were not significantly different (Fig. 7j). These results indicate that the smaller eyes of cave planarians are explained primarily by a lower rate of eye cell production rather than by a higher rate of eye cell death.

**Slower eye nucleation following eye resection in cave planarians**
To further assess the hypothesis that cave planarians have limited numbers of eye progenitors compared to surface-dwelling planarians, impacting eye size, we followed eye regeneration after eye-specific resection and head decapitation in *G. multidiverticulata* (discernible eye morphotype) and *G. dorotocephala*. Major injury in planarians is known to cause a large increase in neoblast proliferation[80–82]. By contrast, small injury, including eye resection, does not induce a sustained increased rate of neoblast proliferation, with eye regeneration resulting from homeostatic replacement of eye cells from constitutively produced eye progenitors[23].

Eye resection resulted in poor to no eye formation up to 18 days later in *G. multidiverticulata* (assessed in the discernible eye morphotype), whereas *G. dorotocephala* eyes had fully formed by this time (Fig. 8a, c). We therefore examined an exceptionally late time point (56 days) following eye resection. Even at this time, some eyes in *G. multidiverticulata* (n = 4) had a lower number of photoreceptor cells than expected, presenting an average of 6 photoreceptor cells per eye, whereas the majority of the animals (n = 8) presented an average of 12

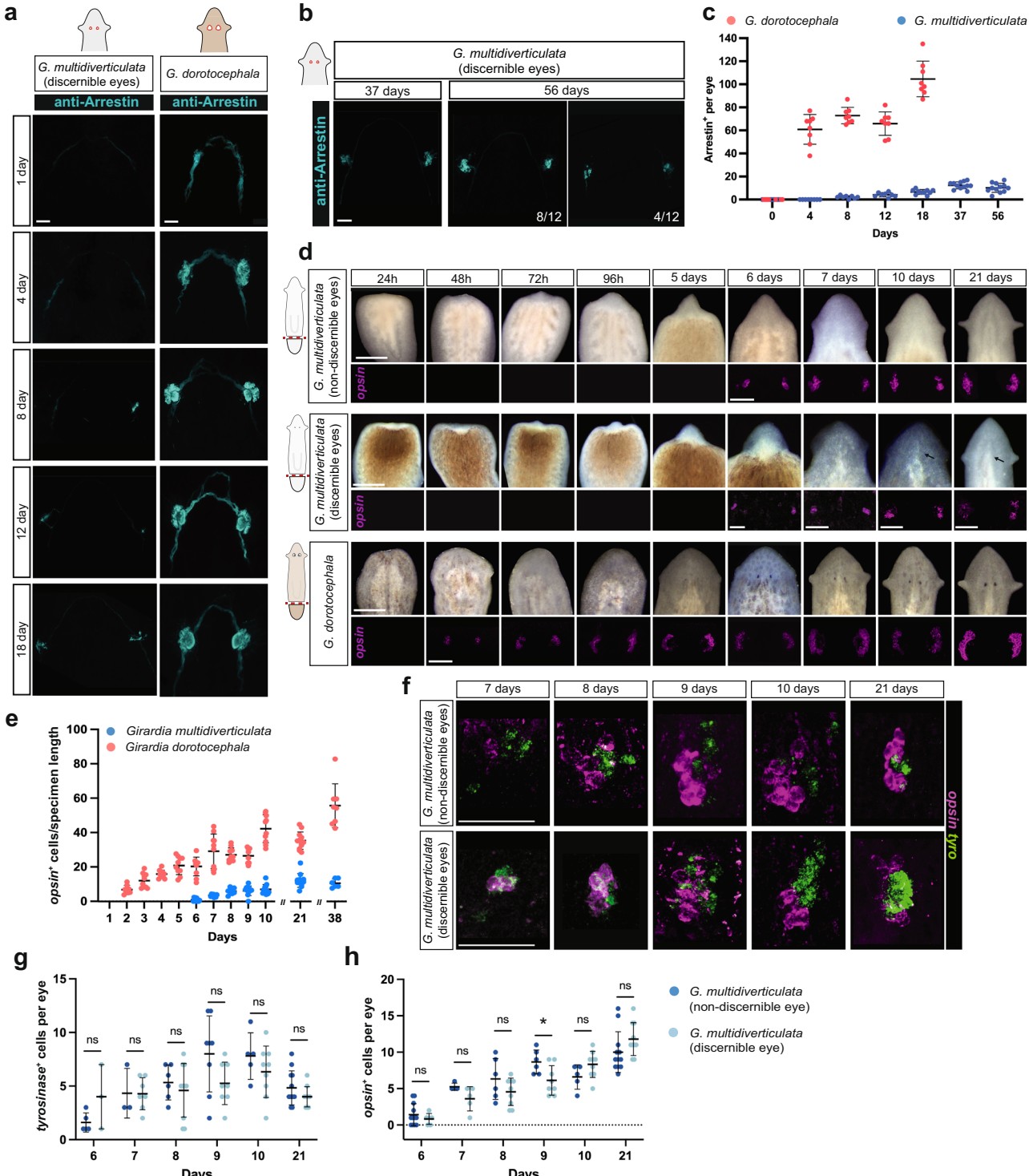

**Fig. 8 | Slower eye regeneration in cave planarians. a** Cave planarians start to nucleate their eyes eight days after eye resections, whereas surface planarians take a shorter time. **b** Even after 56 days eyes cave planarians still presented a lower number of photoreceptor cells than expected. **c** Quantification of PRN (Arrestin⁺) during regeneration after eye resection in *G. dorotocephala* (*n* = 8 for all data points) and *G. multidiverticulata* (*n* = 8, for 0, 4, 8, and 12 days; *n* = 10, for 18 days; *n* = 12, for 37 and 56 days). **d** Blastema formation and eye regeneration in blastemas of surface and cave planarians (*G. multidiverticulata* discernible and non-discernible morphotypes). Live images of regeneration show the differentiation of anterior structures, including eyes and auricles (above); and fluorescent in situ hybridization (FISH) of photoreceptor cells (*opsin⁺* cells) show the early stages of

aggregation and differentiation of eye cells (i.e., nucleation) (below). Note that *opsin⁺* cells only appear after 6 days of regeneration in cave planarians.
**e** Quantification (mean ± SD) of photoreceptor cells (*opsin⁺* normalized by specimen length) during regeneration of *G. dorotocephala* and *G. multidiverticulata*.
**f** Eye nucleation rates are similar between the discernible and non-discernible eye cave planarian morphotypes; quantification is shown in (**g**) and (**h**) with the respective cell quantifications (mean ± SD). Intervals were compared with a Student's two-tailed t-test. ns, not significant, p > 0.05. Scale bars are 1 mm for live images and 50 μm for FISH images. The sample size for each experiment is provided in the "Quantification, statistics, and reproducibility" section.

photoreceptor cell per eye (Fig. 8b, c). The regenerative capacity of cave planarians following resection was therefore limited compared to their surface-dwelling counterparts, likely associated with a lower occurrence of eye-specific progenitors accessible to differentiate into a new eye when compared to surface planarians.

### Slower regeneration following amputation in cave planarians

We next examined whether eye nucleation during head regeneration following transverse amputation also occurred at a slower rate in cave planarians when compared with surface planarians. Head regeneration in planarians occurs in a blastema, which is a regenerative outgrowth formed at amputation planes. *G. multidiverticulata* head blastemas were first observed ~48–72 hpa and were notably smaller than *G. dorotocephala* head blastemas over the first 5–6 days of regeneration (Fig. 8d). Cave planarians (both discernible and non-discernible morphotypes) nucleated eye cells only at around 6–7 days post-amputation in head blastemas, whereas *G. dorotocephala* differentiated eye cells around 48 h post-amputation (Fig. 8d, e), similar to what was previously described for other surface species[18,36]. During the early stages of regeneration, it was challenging to visualize the pigment cells for the discernible eye morphotype, as they only become visible approximately 10 days into the regeneration process (Fig. 8d). The two cave morphotypes had similar overall head regeneration characteristics, with no overt differences in eye differentiation rates observed (Fig. 8f–h). Consistent with the slower overall rate of blastema growth in cave planarians, all organs, including the eyes, brain, and pharynx, also displayed significantly delayed regeneration following amputation (Fig. 9a–d).

We labeled and counted mitotic cells using an antibody that recognizes phosphorylated histone-H3-Ser10 (H3P) at 0, 6, 24, 48, 72, and 96 h post-amputation and in uninjured animals. Uninjured individuals and those examined at the 0-h time point exhibited comparable numbers of mitotic cells between *G. multidiverticulata* and *G. dorotocephala* species, indicating that their mitotic activity is similar under homeostatic conditions (Fig. 9e, f). After tail amputation, the anterior wound epithelium closes and a first generic cell proliferation response to injury spreads through the body by approximately 6 h after amputation[80]. *G. multidiverticulata* also exhibited an initial mitotic peak at 6 h post-amputation, but with significantly reduced cell numbers compared to *G. dorotocephala*. In *Schmidtea mediterranea* after 24 h a second proliferative peak begins, reaching its maximum around 48–72 h after amputation[80,81]. At this stage mitotic cell density is increased close to the wound site and is a component of the missing tissue response. Proliferation below the wound epithelium produces the non-pigmented blastema outgrowth, which will replace the most distal structures. The number of mitotic cells in *G. dorotocephala* was elevated at 24 h post-amputation and remained elevated over the next several days (Fig. 9f). Cave planarians failed to show robust elevation of mitosis following injury, compared to the surface species (Fig. 9f). The second mitotic peak during planarian head regeneration helps in the rapid differentiation of new tissue at the wound site[80]. The reduced number of mitotic cells during the second peak of regeneration likely accounts for the slower rate of formation of brain, pharynx, and eye cells observed in the cave species (Figs. 8d, e; 9a–d). Despite lower mitotic cell numbers in cave planarians during the initial growth phase of the blastema, the head, brain, and pharynx (unlike the eyes) grew and reached similar body proportions to that of the surface species (Fig. 9a–d). Furthermore, mitotic cell counts were comparable in cave and surface planarians under homeostatic conditions, which ultimately determines final organ size (Fig. 9e, f). Prior work in *S. mediterranea* showed that a block of the proliferative increase of the missing tissue response (achieved with RNAi of *follistatin*) still allowed regeneration; and that despite a slower rate of regeneration overall, tissues and organs could nonetheless reach a normal final size[83].

The evolutionary basis for the overall lower rate of regeneration observed in the cave species is unclear, but possible reasons could be considered. A lower rate of predation injuries in the cave environment or loss of an asexual mode of reproduction through transverse fission, common in surface planarians, could have relaxed selective pressure to maintain the heightened proliferative response to major injury. Regardless, the eyes of cave planarians failed to match the same size as their surface counterparts by the end of regeneration, implying differential resource allocation toward the production of new eye cells versus brain and pharynx cells.

## Discussion

A number of Spiralian species display troglobitic traits, but the mechanisms underlying the evolution of these traits are poorly understood. We investigated eye reduction in the planarian *Girardia multidiverticulata* as a new model to study the evolution of eye visual systems in cave animals. *G. multidiverticulata* retained functional, small eyes, and the expression of transcription factors associated with planarian eye formation was largely conserved (with a few variations in cell-type expression). Evolutionary trade-offs are important for the maintenance or loss of evolutionary traits. We found that investment in *G. multidiverticulata* is allocated differently in the production of eye versus other tissues (brain, pharynx, and epidermis) between cave and surface planarians (Fig. 10). Photosensory systems confer advantages, such as for finding food and escaping predation. However, eye development and maintenance in some species can be energetically costly[5,84,85]. Planarian eyes are important for escaping light and finding shelter[40]. In dark environments, eyes in some species can confer no clear adaptive advantage. Selection could favor the maximization of fitness benefits against energetic costs, leading to eye reduction and loss[2,86,87]. In addition, or alternatively, the relaxation of purifying selection could have led to eye reduction in cave planarians. The cave planarian *G. multidiverticulata* presents an interesting case of retaining eye cell types and function despite living in constantly dark surroundings. This either suggests a cost–benefit equilibrium within a fitness peak on an adaptive landscape or could represent an evolutionary transition phase toward complete eye loss. The occurrence of eye morphotypes (discernible and non-discernible eyes) with heritable eye differences, and with both displaying reduced numbers of eye cells, is consistent with the possibility of regressive evolution ongoing in this population.

Fewer eye-specialized neoblasts were present in uninjured *G. multidiverticulata* when compared to surface species. By contrast, *G. multidiverticulata* produced presumptive fate-specified *foxA*+ neoblasts for the pharynx and *agat-3*+ epidermis progenitors in similar numbers to surface planarians. Furthermore, there was a reduced number of newly differentiated eye cells produced from neoblast stem cells per unit time compared to the surface species, but similar production of differentiated *ppl-1*+ and *pkdl-2*+ brain cells per unit time. These findings suggest that a lower rate of specification events toward the eye fate in the proliferative adult stem cells in cave planarians explains their small eyes (Fig. 10).

Eye cell proliferation during embryonic development and adult homeostasis of cave animals has been also addressed in vertebrate models. Cavefish eyes exhibit ongoing stem cell proliferation in the retina ciliary marginal zone[88]. In *Astyanax mexicanus*, the cave morphotype shows constant cell proliferation in the ciliary marginal zone during embryonic and adult development of the retina, similar to the surface morphotype. Eye loss in this species is not caused by a decrease in eye progenitor cell proliferation[89,90]. By contrast, eye reduction in *Sinocyclocheilus anophthalmus* is associated with decreased proliferating cells in the ciliary marginal zone of adult retinas. Whereas *A. mexicanus* experienced cell death resulting in more vestigial eyes, *S. anophthalmus* retained intact eye structures despite their reduction1[91], These results raise the possibility that lower

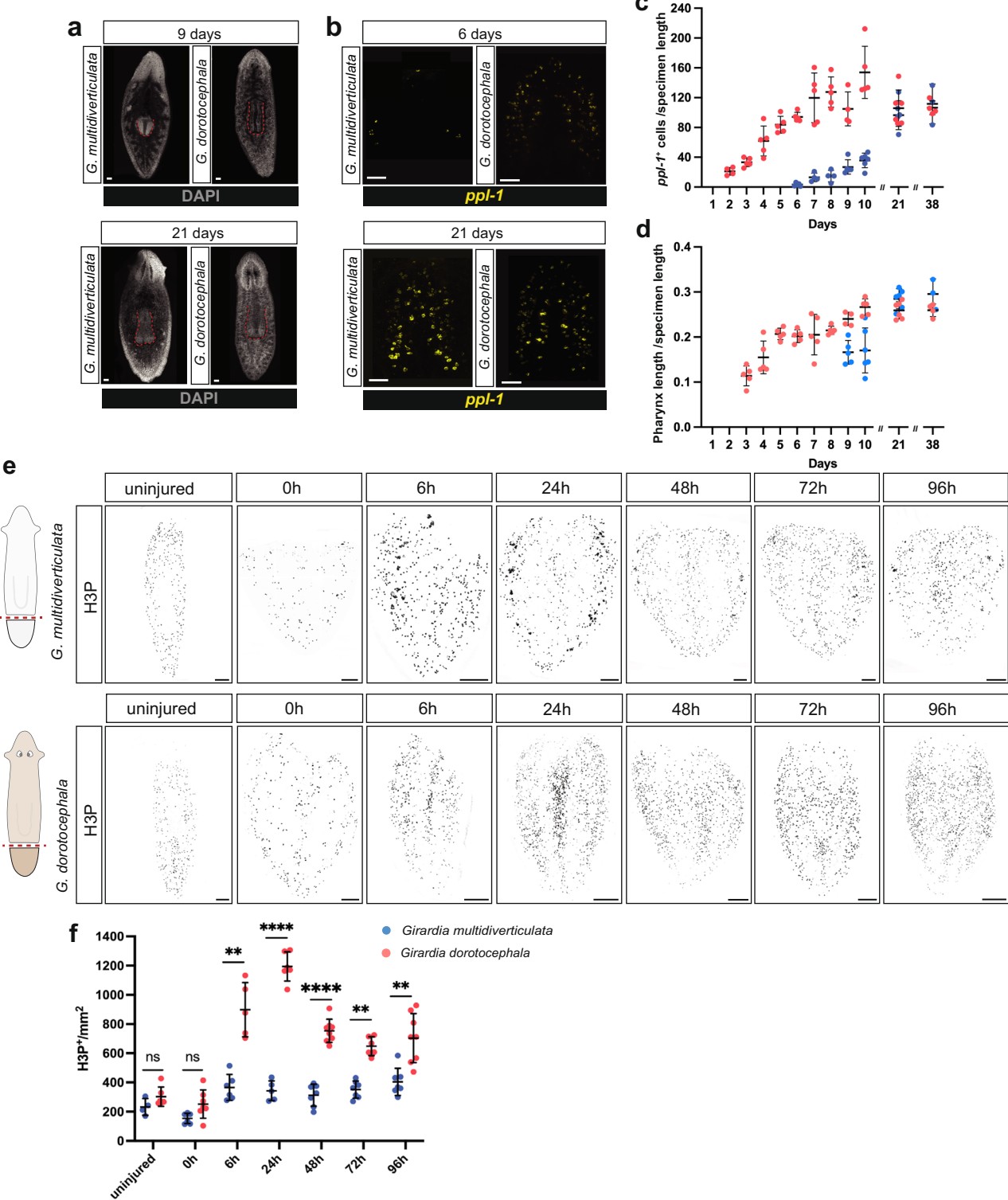

**Fig. 9 | Slower regeneration in cave planarians. a** *G. multidiverticulata* takes longer to regenerate its pharynx (visualized using DAPI); however, after 21 days, structures reach similar sizes. Scale bars, 50 μm. **b** *G. multidiverticulata* takes longer to regenerate its brain (marked with *ppl-1*); however, after 21 days both structures reach similar sizes. Scale bars, 50 μm. **c** Quantification (mean ± SD) of pharynx length (DAPI) normalized by specimen length, during regeneration in *G. dorotocephala* and *G. multidiverticulata*. **d** Quantification (mean ± SD) of brain cells (*ppl-1*[+] normalized by specimen length), during regeneration of *G. dorotocephala* and *G. multidiverticulata*. The sample size for each experiment is provided in the "Quantification, statistics, and reproducibility" section. **e** Cave planarians failed to show robust elevation of mitosis following injury. Immunolabeling of mitotic cells (H3P[+]) in *G. multidiverticulata* non-discernible eye morphotype and in *G. dorotocephala*. Scale bars, 100 μm. **f** *G. multidiverticulata* exhibits lower numbers of mitotic cells (H3P[+]) per mm[2], exclusively during regeneration, when compared with *G. dorotocephala*. H3P[+] cells from each time point were counted, and compared between the two species using a Student's two-tailed t-test; ns, not significant,**p < 0.01, ***p < 0.001, ****p < 0.0001. Mean ± SD. The *G. dorotocephala* counts involved n = 5, n = 7, n = 5, n = 6, n = 8, n = 6, and n = 8 animals for uninjured, 0 h, 6 h, 24 h, 48 h, 72 h, and 96 h, respectively. The *G. dorotocephala* counts involved n = 4, n = 6, n = 6, n = 5, n = 7, n = 7, and n = 7 animals for uninjured, 0 h, 6 h, 24 h, 48 h, 72 h, and 96 h, respectively.

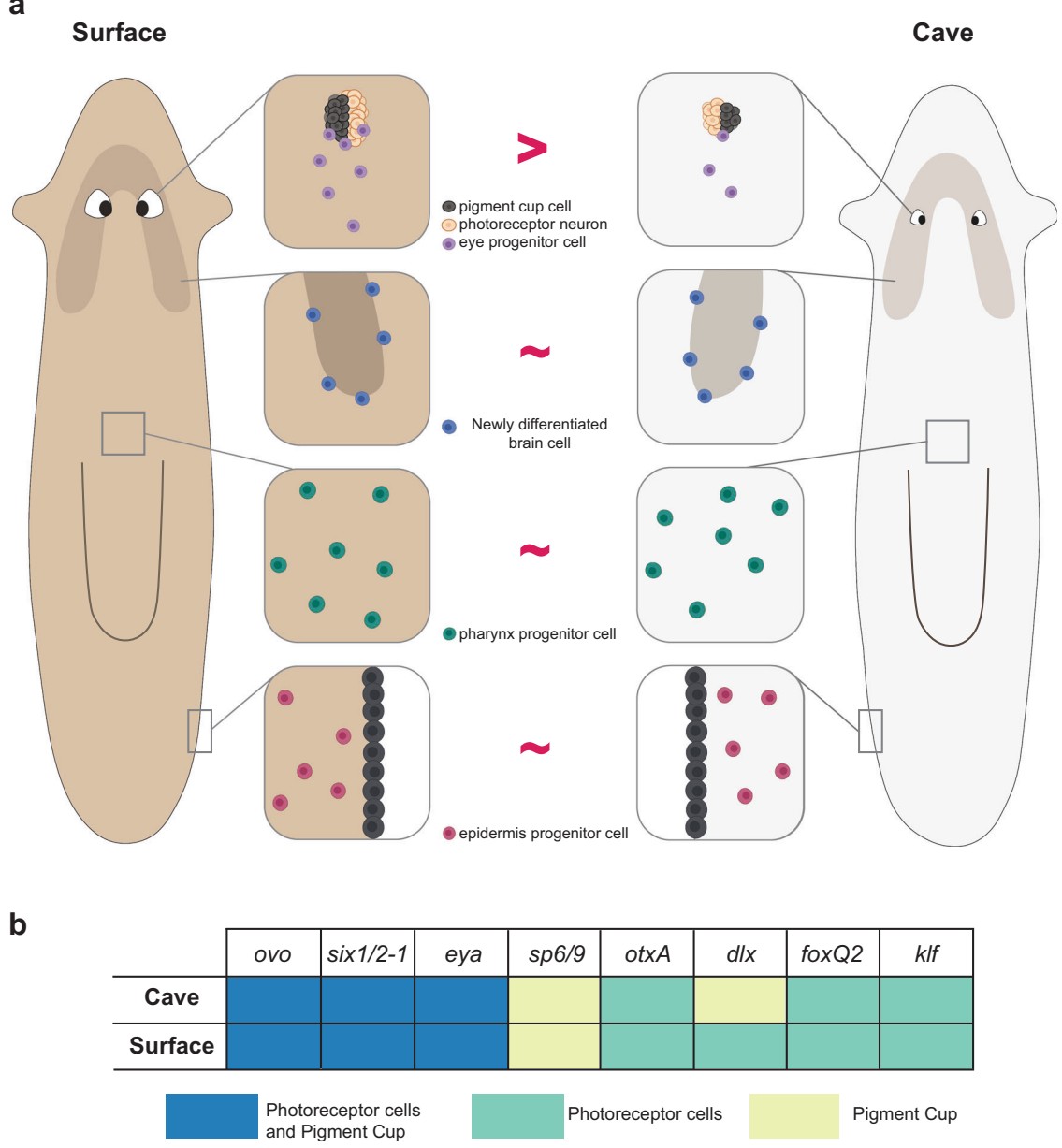

a
Surface — Cave

- pigment cup cell
- photoreceptor neuron
- eye progenitor cell

Newly differentiated brain cell

pharynx progenitor cell

epidermis progenitor cell

b

| | ovo | six1/2-1 | eya | sp6/9 | otxA | dlx | foxQ2 | klf |
|---|---|---|---|---|---|---|---|---|
| Cave | | | | | | | | |
| Surface | | | | | | | | |

Photoreceptor cells and Pigment Cup · Photoreceptor cells · Pigment Cup

**Fig. 10 | Model for eye reduction in cave planarians. a** Changes in eye fate specification rate in adult stem cells in cave planarian evolution resulted in organ size reduction. Brain, pharynx and epidermis progenitor production rates remain comparable between cave and surface species. Created in BioRender. Saad, L. (2024) BioRender.com/f28z409. **b** Illustration summarizing eye transcription factor expression dynamics in cave and surface planarians (indicating a largely conserved program of eye formation, and a difference in expression of *dlx*, in the photoreceptor cells in *G. multidiverticulata*).

production of eye cells from adult progenitors might have independently evolved in fish and planarians.

In *G. multidiverticulata* the gene expression levels of *ovo* and other genes encoding transcription factors important for eye formation showed similar levels per eye cell compared to surface planarians. RNAi of these genes in the cave species resulted in defects in eye formation, similar to the case of surface planarians. This indicates that reduction in cave planarian eye size might arise at an early stage – in neoblasts choosing to activate the eye program or not, rather than by the expression levels of eye-associated transcription factors in cells that do choose an eye fate and differentiate. The regulation of specialized progenitor production rates in planarians is not well understood. Some genes, such as those from the Hippo signaling pathway and epidermal growth factor receptor pathway, are known to control stem cell proliferation and disrupt allometric scaling by altering neoblast production[92–94,95]. Inhibition of *egfr-4* increases the number of eye progenitor cells at the expense of differentiated eye cells, resulting in smaller eyes[92]. Inhibition of a component of the NuRD complex in *Schmidtea mediterranea* increases *ovo*+ cell numbers, resulting in a substantial reduction in eye pigmentation and an increase in eye photoreceptor cells[96]. These findings collectively indicate that modulation of a variety of genes can control alterations in the production of eye stem cells and result in modifications to planarian eyes, but these known cases also cause abnormalities in the overall structure of the organism. In the cave-dwelling planarian *Girardia multidiverticulata*, evolution found a mechanism to adjust eye progenitor production without affecting the function of other organs. Changed rates of component production in a system undergoing turnover present a possible path for evolution to change proportions[97]. This type of mechanism could potentially contribute to changes in overall

body proportions during evolution in multiple contexts, providing a strategy for allometric modification.

Phenotypes related to adaptations to living in the dark among evolutionarily distant cave species are often accompanied by changes at the molecular level[98,99]. For instance, *G. multidiverticulata* and other cave-dwelling species such as salamanders[100], crustaceans[101–104], and various fish species[43,44,48,91,105] display downregulation of genes related to phototransduction –such as *rhodopsin*, *opsin*, and/or *arrestin*. This suggests a case of evolutionary convergence involving similar changes in different lineages of cave-dwelling animals.

In conclusion, we present a mechanistic study of trait reduction in a cave-dwelling Spiralian model. We elucidated a unique evolutionary mechanism that results in reduced organ size involving a lower rate of eye progenitor production from adult stem cells. These findings suggest changes to adult tissue maintenance processes, such as those involving stem cell fate specification, can provide dials for evolution to turn to change the proportions of adult cell types, such as in the case of trait loss in cave-adapted species.

# Methods

## Animal culture and habitat

*Girardia multidiverticulata* were collected from the cave Buraco do Bicho, located in Serra da Bodoquena, Mato Grosso do Sul, Brazil (20°33′50″S and 56°43′50″W) (Fig. 1a), and was first described by Stella Teles de Souza and collaborators[27]. This species is currently classified as Critically Endangered according to the Instituto Chico Mendes de Conservação da Biodiversidade (ICMBio), and a culture of this species is maintained in the Brown Lab in the Department of Zoology at the University of São Paulo.

To contribute to the genetic diversity of the *G. multidiverticulada* population in the laboratory, the cave, Buraco do Bicho, was revisited in 2018, and new specimens were collected and cultured in the laboratory (ICMBio SISBIO permit 93921-1). The surface species *Girardia dorotocephala* (commercially purchased by Carolina Biological Supply Company, Burlington, NC; Item #132970), *Schmidtea mediterranea* strain (CIW4), and *Duguesia japonica* (kindly provided by Agata Lab) were cultured in 1× Montjuic planarian water (1.6 mmol/l NaCl, 1.0 mmol/l CaCl2, 1.0 mmol/l MgSO4, 0.1 mmol/l MgCl2, 0.1 mmol/l KCl and 1.2 mmol/l NaHCO3 prepared in Milli-Q water)[106] at 20 °C. All flatworms were maintained at room temperature, in small tanks, in the dark, and were fed weekly with calf liver. Animals were starved 1–2 weeks prior to experiments.

## Phylogeny analysis

Ribosomal DNA (18S type I and II, and 28S), mitochondrial cytochrome oxidase I (COI), and elongation factor 1-alpha (EF1) genes were used for genetic analyses (Supplementary Table 1). *G. multidiverticulata* gene prediction and annotation were carried out using similarity-based search by blast searching genes from well-annotated closely related Dugesiidae species against the whole *G. multidiverticulata* transcriptome (Supplementary Table 2). Representative sequences for the remaining species were downloaded from GenBank (https://www.ncbi.nlm.nih.gov/genbank/) (Supplementary Table 1).

Nuclear ribosomal markers were aligned with MAFFT using the FFT-NS-2 algorithm[107] and checked by using Geneious 8.1.7 software[108]. For protein-coding COI sequence alignments, we used the TranslatorX pipeline[109]. Nucleotide sequences were translated into amino acid sequences (Translation Table 9), followed by MAFFT alignment, using the FFT-NS-2 algorithm, and then back-translated to nucleotide sequences. EF1 sequences were also translated into amino acids using Geneious 8.1.7 software, aligned with MAFFT (FFT-NS-2), and back-translated to nucleotide sequences. For all four gene alignments, regions of ambiguity were removed using the software Gblocks[110] setting all the options for a less stringent selection. Trimmed sequences were finally concatenated using Geneious 8.1.7 software.

Maximum likelihood (ML) analysis was performed with concatenated datasets in IQTree v1.6.4. The best-fitting model was determined using the ModelFinder algorithm implemented in IQTree[111,112]. We performed 1000 ultrafast bootstrap replicates in IQTree using the TVM + F + I + G4 model. Bayesian Inference (BI) analysis was performed in Mr. Bayes v3.2[113] using 1,110,000 generations, and 25% burn-in was used under the GTR + I + G model. Finally, BI and ML trees and posterior probabilities were visualized using Figtree v1.4.4.

## RNA extraction, library preparation, and sequencing

Intact heads, whole animals, and regenerating head and tail fragments from *G. multidiverticulata* (both discernible and non-discernible eye morphotypes) from 0 h, 6 h, 18 h, 3 days, 5 days, and 8 days were polled and used for RNA extraction. *G. multidiverticulata* fragments from discernible and non-discernible-eyed animals were sequenced separately. Total RNA was extracted using TRIzol (Invitrogen) according to the manufacturer's instructions. cDNA-sequencing library was synthesized using stranded KAPA mRNA HyperPrep (Roche) and Kapa Dual-Indexed Adapter kit Illumina Platforms following the manufacturer's protocol. Libraries were sequenced on an Illumina HiSeq 3000/4000 with 150 paired-end reads for an average sequencing depth of 20 million reads per sample. Illumina sequencing was performed by Genohub. The first sequencing analysis was used to generate probes for FISH, and the second sequencing analysis was used to perform the differential gene expression analysis described below.

After sequencing, the quality of raw reads was accessed using FastQC (http://www.bioinformatics.babraham.ac.uk/projects/fastqc/), and de novo transcriptome assembly was performed with Trinity in which rCorrector was used to remove erroneous k-mers from Illumina paired-end reads[114], TrimGalore was used to remove adapters and low-quality bases (Available in: https://www.bioinformatics.babraham.ac.uk/projects/trim_galore/). The trimmed reads were then mapped to the SILVA database (Available at: https://www.arb-silva.de/) to remove unwanted (rRNA reads). Finally, overrepresented sequences were removed using the Python script, RemoveFastqcOverrepSequenceReads.py (Available at: https://github.com/harvardinformatics/TranscriptomeAssemblyTools/blob/master/RemoveFastqcOverrepSequenceReads.py). De novo transcriptome assembly was performed using the Trinity bioinformatics tool[115] with previous normalization of the edited reads. Assembly statistics were computed using the script TrinityStats.pl, contained in the Trinity package. The proportion of reads mapped to the assembly was assessed with Bowtie2[116]. Later, weakly expressed isoforms were removed based on their expression values following the Trinity protocol. Analysis of homology between de novo assembled transcripts and the planarian database was performed by pair-wise comparison using BLAST. Transcript sequences for candidate genes that showed eye expression in a previous in situ hybridization study[18] were used in BLAST analyses, and are reported in Supplementary Table 4.

## Single-eye purification protocol

For differential gene expression analysis, the eyes from *G. multidiverticulata* (non-discernible and discernible eye morphotypes), *G. dorotocephala*, *S. mediterranea*, and *D. japonica* species were isolated by trimming the surrounding tissues with a microsurgery blade, preserving the general eye structure, with a small amount of surrounding tissues still present. Each isolated eye was then added to 1% beta-mercaptoethanol in TCL buffer (Qiagen 1031576) and was placed at −80 °C (Supplementary Fig. 4d)

## Single-eye RNA extraction

RNA-sequencing libraries from eyes were prepared using a protocol for small amounts of RNA input, as previously described[117]. Each sample was washed using Ampure XP beads (Agencourt). Samples were

then eluted in a solution containing reverse-transcription primer (5′-AAGCAGTGGTATCAACGCAGAGTACT(30)VN-3′, IDT DNA), dNTP, SUPERase RNase-inhibitor (40 U/µl; Life Technologies #AM2696) and water. 7 µl of a solution containing water, 5× Maxima reverse-transcription buffer (Thermo-Fischer), MgCl2, Betaine (5 M; Sigma-Aldrich; B0300-5VL), SUPERase RNase-inhibitor (40 U/l), Maxima RNase H-RT (200 U/µL; Thermo-Fischer, EP0753), and the template switching-oligo (Exiqon; 100 µM; AAGCAGTGGTATCAACGCA-GAGTACrGrG+G; r and "+" denote RNA and LNA bases, respectively) were then added to each sample. Following a PCR reaction, 14 µl of the solution containing water, PCR primers (10 µM; 5′- AAGCAGTGGTAT-CAACGCAGAGT-3′), and KAPA HiFi HotStart ReadyMix (Kapa Biosystems; KK2601) was added to each well. Next, cDNA was amplified, PCR products were purified using ×0.8 Ampure XP beads and were eluted in 20 µl of H$_2$O. Subsequently, the library was synthesized using the Nextera XT Library Kit (Illumina). We prepared separate RNA libraries for each sample, resulting in a total of 12 libraries for *G. multidiverticulata* (discernible), 12 for *G. multidiverticulata* (non-discernible), 11 for *G. dorotocephala*, 11 for *S. mediterranea*, and 13 for *D. japonica*.

## Ortholog prediction
The *G. dorotocephala* transcriptome was assembled with Trinity from RNA-seq data downloaded from SRA (accessions SRR3479045, SRR3479046, SRR3479048, SRR3479052), as described above for the *G. multidiverticulata* transcriptome. The *D. japonica* transcriptome was downloaded from http://www.planarian.jp/seq/DjTrascriptome.fasta.zip. The *S. mediterranea* transcriptome was downloaded from https://planmine.mpibpc.mpg.de/planmine/model/bulkdata/dd_Smed_v6.pcf.contigs.fasta.zip. For 162 genes reported to be expressed in the *S. mediterranea* eye[18], the transcript sequences were manually curated based on multiple sequence alignment between the four species. For all other transcripts, coding sequences were predicted from the transcriptome sequences using Transdecoder. Eye-related orthologs between the four species were then predicted using OrthoFinder. All *G. multidiverticulata* eye genes used in this study can be found in Supplementary Data 1.

## Read mapping and annotation
Smart-Seq2 reads from *G. multidiverticulata*, *G. dorotocephala*, *S. mediterranea*, and *D. japonica* were mapped to their respective transcriptomes with STAR (v2.7.10a). Per-gene read counts were calculated using the R package Rsubread. Cross-species comparisons were made using the set of one-to-one orthologs (8088 genes for the *G. multidiverticulata* – *G. dorotocephala* comparison; and 3088 genes for the four-species comparison). For single-eye samples, the median of ratios method implemented in the R package DESeq2 was used to estimate per-sample size factors and to perform count normalization.

## Differential expression analysis
The R package DESeq2 was used to model read counts by a negative binomial distribution (using the size factors calculated as described above) and to perform hypothesis testing. For single-eye pair-wise comparisons between each cave and surface species, p-values were calculated by the default DESeq2 Wald test. We first evaluated the possible differences between the *Girardia multidiverticulata* discernible eye morphotype pair-wise with all other surface species individually. Then we repeated the same analysis, but using the *Girardia multidiverticulata* non-discernible eye morphotype instead (2 morphotypes × 3 surface species = 6 pair-wise combinations) Supplementary Table 3. Next, we assessed the genes that were differentially expressed in both cave planarian morphotypes when compared with all other surface planarians.

Data analyses included: (1) violin plots displaying the average expression levels of eye-related genes across the different species based on log$_2$-normalized counts; (2) a differential gene expression table comparing the data from single eyes between each cave and surface species, as well as between the two cave morphotypes, using pair-wise DESeq2 Wald tests (Supplementary Table 3); and (3) heatmaps of only the differentially expressed genes found in common between the two cave planarians morphotypes and surface planarians, considering those with an adjusted p-value less than 0.05 and a log$_2$FoldChange > 1, log$_2$FoldChange < −1. Heatmaps were constructed using Pheatmap.

## Gene cloning
Specific primer sequences for each target gene were used with Gateway adapters or the addition of the T7 promoter sequence (Supplementary Table 4). Primers were generated using Primer3[118,119]. Genes were cloned into pGEM (pGEM T-Easy, Promega) for use in riboprobe and dsRNA reactions. The resulting recombinant plasmids were transformed into competent DH10B cells (Thermo Scientific) and grown in overnight culture. Plasmid DNA from colonies was purified using the QIAprep Spin Miniprep Kit (Qiagen). Plasmids were then sequenced by Sanger sequencing (GENEWIZ). After confirmation, in vitro transcription reactions were performed with T7 (Promega), and the product was used to generate DIG-, and FITC- (Roche) labeled ribonucleotides. RNA was purified using ethanol precipitation with 7.5 M ammonium acetate. Pellets were resuspended in formamide and were stored at −20 °C.

## Fluorescence in situ hybridization (FISH)
FISH followed the methodology previously described in refs. 120,121 with modification as described below. Sequences used for all FISH probes are provided in Supplementary Table 4. In some cases, the same probe could be used for both *G. multidiverticulata* and *G. dorotocephala*, because of sequence similarities. For *G. dorotocephala* probes, transcripts available in NCBI were used (https://www.ncbi.nlm.nih.gov/ PRJNA316134). Animals were killed and mucus was removed in 5% NAC for 3 min. Fixation was performed in 4% formaldehyde in PBST for 15 min. Because of *G. multidiverticulata* tissue fragility, we stored worms in mesh baskets to perform the following procedures. Animals were rehydrated and bleached for 90 min at room temperature. Animals were then treated with 2 µg/ml proteinase K and were incubated in a prehybridization solution (50% formamide, 5× SSC, 1 mg/ml yeast RNA, 1% Tween-20) for 2 h at 56 °C. Next, worms were hybridized with RNA probes diluted at 1:800 overnight. On the next day, worms were washed two times with prehybridization, 1:1 prehybridization solution: 2× SSC, 2× SSC:PBST, 0.2× SSC:PBST at 56 °C. Subsequently, specimens were incubated in blocking solutions for 90 min at room temperature prior to labeling with anti-DIG-POD (1:1500, Roche #11207733910), anti-FITC-POD (1:2000, Roche #11426346910) respectively. On the next day, anti-DIG-POD or ant-FITC-POD were washed and underwent tyramide development. Prior to antibody labeling for a second probe, and/or for immunohistochemistry, peroxidase inactivation was performed in 1% sodium azide. Animals were labeled in a solution of 1µg/ml DAPI (Sigma) prior to mounting on slides.

## In situ Hybridization Chain Reaction (HCR)
Probes for *ovo*, *foxA*, and *sp6-9* HCR FISH were designed in OligoMiner (https://doi.org/10.1073/pnas.1714530115) (https://github.com/beliveau-lab/OligoMiner), and through custom Python and R scripts (https://github.com/cooketho/make_hcr_probes) for each target transcript. The full sequence of each probe, including initiator and spacer sequences, can be found in Supplementary Table 5. Probes were ordered as oligo pools from IDT (50 pmol/oligo) and were resuspended in water to 1 µM.

Animals were killed, fixed, rehydrated, and bleached as described above for the FISH protocol. After bleaching, the animals were washed in PBST and a methodology for the HCR FISH protocol described in

ref. [122] was performed. The probe concentration was 1.6 μl of the probe from 1 μM stock solution.

## Immunohistochemistry

The animals were killed, fixed, bleached, and treated with proteinase K as previously described here for FISH. For the anti-Arrestin protocol, the animals were placed in blocking solution (10% horse serum in PBSTx) and were labeled with mouse anti-Arrestin (1:5000) (kindly provided by Kiyokazu Agata) in blocking solution, overnight at 4 °C. Samples were then developed with fluorescein tyramide in borate buffer (1:1500) and were labeled with DAPI prior to mounting.

For anti-H3P immunofluorescence, animals were placed in anti-phospho-Histone-H3 antibody (Millipore 05-817R-I, clone 63-1C-8) overnight at room temperature at a concentration of 1:300 in 5% inactivated horse serum. After PBSTx washes, samples were labeled with a goat anti-rabbit-Alexa 647 secondary antibody (1:300) in a block (5% inactivated horse serum) overnight at room temperature. After PBSTx washes samples were labeled with DAPI prior to mounting.

## F-ara-EdU immunofluorescence

For F-ara-EdU (EdU) labeling, size-matched *G. multidiverticulata* and *G. dorotocephala* were soaked for 24 h in 1 mL planarian water including 6.25 μL of 200mg/mL F-ara-EdU (T511293 Sigma). After soaking in EdU, animals were kept in 1:1 Montjuic planarian water: Instant Ocean (5 g/L) mixture for 4, 8, and 12 days. Prior to F-ara-EdU immunofluorescence, FISH was performed as described above. After inactivation in 1% sodium azide solution and a series of six wash steps, animals were placed in a Click reaction solution (78.9 μl PBS, 1 μl 100 mM CuSO4, 0.1 μl 10 mM azide-flour 488 Sigma 760765, 20 μl 5 0 mM ascorbic acid) and were incubated in the dark for 30 min at room temperature. Following additional PBSTx washes, animals were labeled with DAPI prior to mounting. F-ara-EdU immunofluorescence protocol was adapted from previous work[123].

## RNAi

Double-stranded RNA (dsRNA) was synthesized as described before[124]: PCR-generated templates of sequences for the forward and reverse of target genes were prepared with a 5′ flanking T7 promoter (TAA-TACGACTCACTATAGGG) (Promega). Then, forward and reverse templates were mixed in separate reactions with 10 mM rNTPs (Promega), 1 M dithiothreitol (DTT; Promega), 5× Transcription Buffer (Promega), and T7 polymerase. Reactions were then incubated overnight at 37 °C. Next, forward and reverse strands were combined and the solution was mixed with 3 M Sodium Acetate and followed by ethanol precipitation. Samples were then resuspended in 25 μl of Milli-Q H₂O. For feeding, 12 μl of dsRNA was mixed with 28 μl of 100% homogenized calf liver and 2 μl of food dye. Animals were starved for at least 10 days prior to the first feeding. Worms were fed with liver containing dsRNA every three or four days, for at least 14 days. Regenerating animals were fixed 14 days after the cut. Animal feeding was evaluated by the red coloring of the gut branches. The *Caenorhabditis elegans unc-22* gene[125] was used as negative control dsRNA. Eyes abnormality was observed in several RNAi conditions. The presence of an abnormal phenotype was determined by blind scoring. We obtained multiple eye images for each replicate of each experimental group (RNAi of control, *six1/2*, *dlx*, *ovo*, *otxA*, and *foxQ2*). These images were then blindly evaluated by two independent reviewers, who classified the eye images as either normal or abnormal. Each picture analyzed was classified as either normal or abnormal by two blinded examiners. These results were then compared to random predictions using Fisher's exact test (p < 0.01). Significant differences between proportions indicated eye abnormalities.

## Eye cell number and brain size comparison

Differently sized animals from *G. multidiverticulata*, *G. dorotocephala*, *S. mediterranea*, and *D. japonica* were selected to perform the experiment. The length of each animal was measured from the tip of the head to the end of the body in a fully stretched-out state. Measurements were performed using AxioVision software. After size acquisition, each animal was allocated to a different well in a 24-well plate, and FISH and anti-Arrestin immunostaining were performed as described previously. For each animal, photoreceptor and/or *ppl-1*-positive cells were manually counted in blind-scored images.

## Behavior

Light intensities for behavior experiments were performed according to a previous study[26]. Two different arenas were used; one presented 12 different light intensities from darker to lighter (TA, test arena), and the other with only one light spectrum (CA, control arena) (Supplementary Fig. 3). Each arena was generated using an iPad as a surface, presenting a continuous display. A rectangular plate (12.5 cm × 8.5 cm × 1 cm) containing 0.5 cm height of Montjuic planarian water was placed on top of the iPad, which was covered to eliminate any other external light from the test environment. An iPhone was placed on top of the box to record videos of behaving animals under different test conditions. A total of 5 animals per testing group were placed in the middle of the arena within the boundaries of the 5th and the 6th bands in the arena at the beginning of each trial. Each trial was repeated 10 times, totaling 50 animals. Animals were recorded for a total of 5 min, and the positions of each animal at the end of each minute were annotated. Statistical analysis was performed by the comparison between the average position of all animals at the end of each minute with an average position of 6 (indicating random distribution). Both eyes were resected in the negative control group.

For eye resections, animals were placed on a moist filter paper on a Peltier plate in order to limit movement, and the tip of a microsurgery blade was used to remove eyes, following the methodology previously discribed[26]. For resections on the *G. multidiverticula* non-discernible eye morphotype, the region where eyes are located in the discernible-eyed morphotype was used as a reference. Animals were used for behavior experiments one day after eye resection. A total of 22 animals were used for the eye resection and behavior experiments.

To assess behavior in different wavelengths, plastic optical filters were used that emit different light wavelengths (Rainbow Symphony Store #10026); filters were placed on top of an iPad displaying an arena with a light gradient. We used red (~625 nm), green (~517 nm), and blue (~465 nm) filters to assess the responses in different wavelengths. A rectangular plate (12.5 cm × 8.5 cm × 1 cm) containing 0.5 cm height of Montjuic planarian water was placed on top of the filters, and behavior was recorded as described previously (Supplementary Fig. 3). A total of 5 animals per test group were placed in the middle of the arena, each trial was repeated 5 times, totaling 25 tested animals per species. Statistical analysis was performed as described above.

## Irradiation

Animals were irradiated using a dual Gammacell-40 137cesium source to deliver 6000 rads. Following irradiation, animals were kept in Montjuic planarian water supplemented with gentamicin (100 mg/mL gentamicin sulfate) and were fixed at 0, 8, and 12 days after irradiation. Later, animals were used in FISH, and *opsin*+ cells were manually counted for each animal. For calculating the exponential decay in eye cell numbers, eye cell counts were log-transformed, and the simple linear regression slopes were compared between *G. multidiverticula* and *G. dorotocephala* using GraphPad Prism software (GraphPad Inc., La Jolla, CA).

## Quantification, statistics, and reproducibility

Live images were obtained using a Zeiss Discovery microscope and an AxioCam camera. The two cave morphotypes were differentiated by the presence and absence of eye pigmentation when observed in a routine Stereo microscope under white light. Specimens that

presented eye pigmentation were categorized as "discernible eye morphotypes", whereas the "non-discernible eye morphotype" exhibited no distinguishable visible pigmentation in their eyes. Fluorescence images were acquired using a Leica SP8 or a Leica STELLARIS 5 confocal microscopes. ImageJ software was utilized for image processing and quantitative analyses.

All cell counts, except for H3P counts (which used semi-automated counting, see below, Fig. 9) were quantified blindly by one observer. Positive cells were called if their fluorescence signal was co-localized with a DAPI-positive nucleus and could be clearly distinguished from background levels. Positive cells were counted through the whole z-stack of the animal, from the dorsal to the ventral epidermis. For all experiments, the optical section was consistently set to the maximum value during z-stack imaging to ensure comprehensive data capture. This approach ensured that each optical plane overlapped with the adjacent planes, thereby preventing the exclusion of any cells during the imaging process. For all experiments measurements were taken from distinct samples.

*FoxA* and *agat-3* cell counts were performed in animals of similar size. Counts were considered positive if fluorescence signals were co-localized with DAPI through the whole z-stack encompassing that cell. For *FoxA* counts a fixed area of 300 μm wide × 200 μm high, located at the parapharyngeal region, was scored. This area excluded the pharynx itself from the counts. For *agat-3* cell counts a fixed area of 350 μm wide × 200 μm high located below the eyes at the auricles region was scored.

Cell counts were performed at 4, 8, and 12 days post-EdU delivery (Supplementary Fig. 6a). For the 4-day time point, cells were counted at 63× confocal magnification. For the 8- and 12-day time points, cells were counted at 25× magnification using confocal microscopy. In all cases, a fixed 100 × 100 μm square region was analyzed within the animal. For the 4-day time point, the imaged area spanned from the anterior pharynx opening to the connection between the dorsal and ventral epidermis, with cell counts focused on the region closest to this epidermal boundary, excluding the epidermis itself. For 8- and 12-day timepoints the imaged area started at the end of the auricles. EdU$^+$/*pkd1l-2*$^+$ double-positive cells were counted from the maximal extension of the brain until the end of the animal head.

A semi-automated approach employing ImageJ was used to quantify the H3P counts (Fig. 9b). First, we analyzed the cell in a Maximum Intensity Z-projection, stacking all optical planes together. The image threshold was set to match the original data, and watershed segmentation was applied to separate any overlapping objects. Subsequently, the "Analyze Particles" function was utilized (Size pixel: 10-infinity, Circularity: 0.3-1.00). A comparison between manual counting and the semi-automated method revealed a difference of only 5–10 cells out of 200–500 total cells counted, validating the reliability of the approach. Total animal area was measured and the total numbers of H3P$^+$ cells were divided by the total area. *G. multidiverticulata* gonads were excluded from the counting.

The total number of *piwi-1*$^+$ cells was obtained by modeling in 3 dimensions using Imaris (Oxford Instruments) based on a previously published approach[126]. First, images of similar-sized animals labeled for *piwi-1* transcripts and DAPI were obtained with 63× confocal magnification. All images were taken at the same region: just anterior to the pharynx and lateral from the pharynx to the edge of the animal; the entire dorsal-ventral section was obtained in a Z-stack (Fig. 6e). Using Imaris software, DAPI segmentation was generated using native segmentation tools and errors were corrected by deleting, fusing, or fragmenting incorrectly segmented cells. The same segmentation was applied to all images. Positive cells were identified by thresholding and manual review of the FISH signal. Images were quantified blindly by an independent observer. Imaris software was used to process and quantify the multi-z-plane confocal images. Cells were delineated using a DAPI signal and then classified based on FISH signal

thresholding. Any detection errors were manually corrected by reclassifying the segmented surfaces according to the FISH signal. After the cell segmentation, Imaris automatically counted the total number of cells present in the 3D z-stack. The numbers of DAPI$^+$ and *piwi-1*$^+$ cells were obtained, and the percentage of *piwi-1*$^+$ cells was calculated by the number of DAPI$^+$ cells divided by the number of *piwi-1*$^+$ cells.

Counts performed in Fig. 8e for *G. dorotocephala* correspond to $n = 8$, $n = 10$, $n = 10$, $n = 10$, $n = 10$, $n = 10$, $n = 10$, $n = 8$, $n = 10$, $n = 12$, $n = 8$, for 2, 3, 4, 5, 6, 7, 8, 9, 10, 21, and 38 days respectively. For *G. multidiverticulata* counts correspond to $n = 10$, $n = 8$, $n = 8$, $n = 9$, $n = 12$, $n = 12$, $n = 6$, for 6, 7, 8, 9, 10, 21, and 38 days respectively. Counts performed in Fig. 9c for *G. dorotocephala* correspond to $n = 5$, $n = 5$, $n = 5$, $n = 5$, $n = 5$, $n = 5$, $n = 4$, $n = 5$, $n = 6$, $n = 4$, for 3, 4, 5, 6, 7, 8, 9, 10, 21, and 38 days respectively. For *G. multidiverticulata* counts correspond to $n = 5$, $n = 6$, $n = 6$, $n = 3$, for 9, 10, 21, and 38 days respectively. Counts performed in Fig. 9d for *G. dorotocephala* correspond to $n = 4$, $n = 5$, $n = 5$, $n = 5$, $n = 5$, $n = 5$, $n = 5$, $n = 4$, $n = 5$, $n = 6$, $n = 4$, for 2, 3, 4, 5, 6, 7, 8, 9, 10, 21, and 38 days respectively. For *G. multidiverticulata* counts correspond to $n = 5$, $n = 4$, $n = 4$, $n = 5$, $n = 6$, $n = 6$, $n = 3$, for 6, 7, 8, 9, 10, 21, and 38 days respectively. Counts performed in Fig. 8g for *G. multidiverticulata* non-discernible eyes correspond to $n = 5$, $n = 3$, $n = 6$, $n = 8$, $n = 5$, $n = 12$, for 6, 7, 8, 9, 10, and 21 days respectively. For *G. multidiverticulata* discernible eye counts correspond to $n = 3$, $n = 7$, $n = 10$, $n = 8$, $n = 9$, $n = 10$, for 6, 7, 8, 9, 10, and 21 days respectively. Counts performed in Fig. 8h for *G. multidiverticulata* non-discernible eyes correspond to $n = 12$, $n = 4$, $n = 6$, $n = 6$, $n = 5$, $n = 12$, for 6, 7, 8, 9, 10, and 21 days respectively. For *G. multidiverticulata* discernible eye counts correspond to $n = 6$, $n = 5$, $n = 9$, $n = 8$, $n = 9$, $n = 10$, for 6, 7, 8, 9, 10, and 21 days respectively. For Figs. 8e, 9c, d, counts were normalized by animal length because sized matching animals were not used in this case.

All statistical analyses were performed in the GraphPad Prism software package (GraphPad Inc., La Jolla, CA). Statistical tests, significance, data points, error bars, and other information relevant to figures are described and explained in the corresponding legends. Before all statistical analysis, a normality test was performed to indicate the best test parameters to be considered. For parametric data one-way ANOVA test followed by Dunnett's multiple comparison test was used when analyzing more than two conditions and an unpaired Student's t-test was used when comparing two conditions. For non-parametric data, Kruskal–Wallis was used when analyzing more than two conditions and Mann–Whitney was used when comparing two conditions. Exact p-values for each experiment are provided in the Source Data File.

### Reporting summary

Further information on research design is available in the Nature Portfolio Reporting Summary linked to this article.

## Data availability

The FASTA file with all the *G. multidiverticulata* ortholog eye genes generated in this study are provided in the Supplementary Data file. The data generated for Fig. 1c in this study are provided in the Supplementary Fig. 1, Supplementary Table 1,2. The data generated for Fig. 5 in this study are provided in Supplementary Table 3. All RNA-seq data generated in this study have been deposited in National Center for Biotechnology Information under accession codes PRJNA1177453 (SmartSeq reads), PRJNA1177451 (Bulk seq reads), and PRJNA1177450 (Transcriptome assembly reads). Source data are provided with this paper.

## Code availability

HCR Fish probe design was made through custom Python and R scripts available at https://github.com/cooketho/make_hcr_probes. *Girardia*

*multidiverticulata* transcriptome assembly, mRNA-seq of eye cells, and script used are available at https://doi.org/10.5061/dryad.4qrfj6qm0.

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

## Acknowledgements

We thank Lívia Medeiros Cordeiro Borghezan and Eleonora Trajano for donating the initial population of cave planarians; Lívia also helped collect additional animals. We are grateful to members of Reddien lab and Brown lab, especially M. Lucila Scimone for helpful comments on the project and manuscript and Chanyoung Park for helping with 3D image reconstructions. P.W.R. is an Investigator of the Howard Hughes Medical Institute and an associate member of the Broad Institute of Harvard and MIT. P.W.R. acknowledges NIH R35 GM145345 for support. We thank CNPQ (Conselho Nacional de Desenvolvimento Científico e Tecnológico Proc. No. 169053/2017-2, L.O.S), CAPES (Coordenação de Aperfeiçoamento de Pessoal de Nível Superior – Brasil, Código de Financiamento 001, L.O.S), and FAPESP (Fundação de Amparo à Pesquisa do Estado de São Paulo Proc. No. 2018/06418-0, 2015/50164-5 and BEPE 2019/18147-4, L.O.S, F.D.B.) for the financial support.

## Author contributions

L.O.S., P.W.R., F.D.B. designed the study; L.O.S, T. C., K.D.A. carried out experiments; L.O.S, T. C., K.D.A. analyzed data; L.O.S., P.W.R., F.D.B. wrote the manuscript, other authors provided editorial comments; P.W.R., F.D.B. supervised the research.

## Competing interests

The authors declare no competing interests.
