## [Peer Review file · Nature Communications]

Reduced adult stem cell fate specification led to eye reduction in cave planarians

Corresponding Author: Professor Frederico (D) Brown

Version 0:

Reviewer comments:

Reviewer #1

(Remarks to the Author)

The manuscript "Reduced adult stem cell fate specification led to eye reduction in cave planarians" by Saad et al. investigates the mechanisms behind eye loss in the cave planarian species *Girardia multidiverticulata*, focusing on the role of adult stem cells (neoblasts) in this evolutionary adaptation. The study reveals that the reduced eye size in these cave-dwelling planarians is not due to a decrease in overall stem cell numbers but results from a lower rate of stem cell specification to eye progenitors. This finding suggests a novel evolutionary mechanism for trait loss, highlighting the importance of stem cell fate specification in organ size reduction and adaptation to dark environments. I enjoyed reading the paper, it is thoroughly done, well written, and the conclusions are in line with the results. It certainly adds to the knowledge in the field. I have no major concerns, though I would like to know what the author's thoughts are on why this species shows decreased tissue regeneration despite having apparently no difference in stem cell-derived progenitors for tissues other than the eye.

Reviewer #2

(Remarks to the Author)

Eye loss is a common phenomenon in troglobites, but has been largely studied in vertebrates (e.g. fish) and ecdysozoan species (e.g. crustaceans). To see how eye loss has occurred in more distant species (i.e. the spiralian clade), the authors used a species of cave planarians, *Girardia multidiverticulata*, which have reduced eye size. The authors identify two morphotypes of *Girardia multidiverticulata* and find that both maintain small, functional eyespots. The authors also go on to show that much of the genetic program of eye development is shared between *Girardia multidiverticulata* and other planarian species, with an interesting exception of *dlx*. To investigate why *Girardia multidiverticulata* have smaller eyespots, the authors investigated cell birth for eye cells and claim that a lower rate of cell birth explains small eye size.

Strengths of the paper include the identification and broad characterization of eyes in a new species, which required foundation building (e.g. transcriptome creation and adaptation of methods). The work is also thorough, with rigorous quantification and thoughtful data presentation. However, the main claim of slow cell birth specific to eye cells requires more evidence before publication. Specific comments and questions follow.

Major comments or questions:

1. The main mechanistic argument made in this paper is that *Girardia multidiverticulata* have smaller eyespots due to a slower rate of cell birth for eye cells. They also present convincing data that fewer eye cells are born in a given period in this species (Fig. 5A/C). However, the EdU+ cell number in general seems much lower in *Girardia multidiverticulata* compared to *Girardia dorotocephala* (Fig. 5B), raising the possibility that the observation is due to an overall low rate of cell birth rather than a specific difference for the eye.
 - a. Can the authors quantify total number of EdU+ cells/animal at these time points?
 - b. Are the number of stem cells or mitotic stem cells different in the two species (the Smedwi-1 staining in Fig. S5B does look fainter in *Girardia multidiverticulata*).
 - c. Can the authors show that there is no difference in EdU+ cell types in addition to EdU+/ppl-1+ cells (which are present in such low numbers that differences might be obscured)?
 - d. FoxA is present in both progenitors and differentiated cells in *Schmidtea* and some of the differentiated cells are outside of the pharynx (Adler, 2014). Further, it's possible that FoxA or other progenitor markers might mark cells differently in cave

planarians. Can the authors perform double FISH with FoxA and Smedwi-1 (or another progenitor/stem cell marker) to clearly mark pharyngeal progenitors to quantify them? Additionally, the FoxA+ cells counterintuitively look more abundant in *Girardia multidiverticulata* in Fig. 5F, but are shown as not significant in Fig. 5G. Methods for cell counting should be detailed more clearly here and elsewhere.

2. In the manuscript, the authors identified two morphotypes that segregate in near-Mendelian ratios and retain their phenotype across amputations, but there is no follow-up on molecular or cellular differences between these genotypes/phenotypes of planarians.

a. Are there genes that are expressed differentially in the *Girardia multidiverticulata* that have visible or non-visible eyes that would explain the difference? I see some data to this effect in Fig. S3 but it might make sense to put it in the main figure if this is a main point of the work.

b. Are there cellular differences, for example in the number of eye cell birth rates in these two morphotypes?

c. As a more minor comment, it is then a bit unclear about whether Fig. 4-6 are with one morphotype or with all cave planarians taken together.

3. The authors say that *Girardia multidiverticulata* have slower eye nucleation, but also slower regeneration of the head. Are other head markers (anterior markers/other brain markers) also slower to regenerate? If so, then perhaps the eye nucleation may not be a specific phenotype.

4. Does the reduction of photoreceptor neurons in *Girardia multidiverticulata* come with a corresponding decrease in other neurons in the visual circuit within the brain? This might be evident from the RNA-Seq results, focusing on the differentially expressed genes that are not enriched in expression in the eye.

5. Is it possible that *Girardia multidiverticulata* vision is adapted to low light or different wavelengths of light compared to surface species? I wonder if they would perform differently than surface species in a light gradient that is dimmer or if they might be more responsive to long wavelengths of light. I also wonder if *Girardia multidiverticulata* might have retained limited vision to see bioluminescent prey in caves?

6. Some of the methods were unclear. For differential gene expression experiments, it wasn't clear which figures/experiments used which RNA sequencing strategy. Addition of diagrams in the figures would help the reader sort this out much more easily. It also wasn't clear which figures included data from the single eye purification protocol.

Minor comments:

1. Dugesia is misspelled in some of the figures.

2. Occasionally, sample labels are missing from figures (e.g. Fig. 2e, f, g).

3. I appreciate the data presentation in Fig. 2 but the black averages are not always visible in this format. Could these be modified slightly to be clearer?

4. Given that there are multiple SoxB genes in the *Schmidtea* genome, it would be helpful to identify which one this is. Is the SoxB here similar to SoxB in *Lapan* which was renamed SoxB1-1 in Ross (2018)?

5. While I appreciate the complexity of the data shown in Supp. Fig. S3, it is hard as a reader to make meaning of the data with all individual datapoints shown. Averages for each gene with notation of statistical differences might be easier to parse. Likewise, Supp. Fig. S4 lacks notation of whether statistical differences were present between any sample, so it is challenging to draw inferences from the data.

6. Full sequences for all *Girardia multidiverticulata* genes used in this study should be included in a supplemental figure and deposited so that others could replicate

Reviewer #3

(Remarks to the Author)

This manuscript from Saad et al describes a potential new mechanism underlying the small-sized eyes observed in cave-dwelling planarians from Brazil. Borrowing from the molecular and cellular description of eye composition in *S. mediterranea*, the authors characterize the cellular composition of *G. multidiverticulata* eyes, demonstrate that despite their small size, they retain residual function in light aversion, perform a comparative transcriptome analysis of cave versus surface planarian eyes, and describe eye regeneration. Some experiments are quite convincing. For example, the behavioral characterization shows that these rudimentary eyes are still sufficient to mediate light avoidance is clear and includes convincing controls. Other experiments and their conclusions have issues (comments below).

Ultimately the model provided is that stem cells in *G. multidiverticulata* have a decreased probability to adopt a photoreceptor fate. This model leans heavily on what's known in *S. mediterranea*, where extensive characterization of stem cells and their descendants has shown that the plentiful stem cells can adopt a variety of organ specific cell fates, although it remains unclear whether this happens at the level of a single cell or collectively among multiple stem cells. This distinction is muddled in the writing. However, another possibility is that the stem cell population as a whole is programmed to churn out organ specific descendants at a certain rate. This rate seems to be overall slowed down in *G. multidiverticulata* as compared to surface planarians. This is supported by data in Figure 6, and should be addressed.

Another issue with this model is that the authors state that neoblasts 'make the choice' to become an eye progenitor at a lower rate than in surface planarians. This conclusion partly depends on the presence of FoxA cells being equivalent in *G. multidiverticulata* and *G. dorotocephala* (model figure). However, the data used to support this, in particular with respect to FoxA, is incorrect. In *S. mediterranea* it's known that not all FoxA-positive cells are stem cells. To make this claim about FoxA cells being stem cells, either EdU labeling or co-labeling FoxA along with a stem cell marker would be necessary. Otherwise it should be removed.

Below are major and minor comments that will help to clarify the data presentation and provide more evidence regarding stem cell behaviors being responsible for the morphological difference in *G. multidiverticulata*.

1. The data suggests that regeneration in this species occurs more slowly than surface-dwelling planarians, suggesting an alternative possibility which is that stem cells are in general just slower to respond both to injury and death. Basic characterization of stem cell dynamics in the two species should be included, such as basal cell proliferation or differentiation rates (with phosphohistone H3 and/or EdU) and injury-induced proliferation responses.
2. Much of the argument that *G. multidiverticulata* and *G. dorotocephala* differ in their stem cell-specific differentiation rests on evidence that there are key changes occurring at the transcriptional level. The RNA sequencing in Figure 4d shows that on average, there may be a difference, but it appears that the violin plots are strongly skewed by a couple of outlier data points (in ovo, *six1/2*, *dlx*, *foxQ2*). Are these outlier points from the same samples? It is important to address this because on average, *ovo* expression in *G. multidiverticulata* and *G. dorotocephala* is overall quite similar except for these outliers; it's discordant with the *in situ* expression in Figure 5e.
3. *G. multidiverticulata* animals appear to be overall lighter in color than other surface-dwelling planarians, suggesting that differences in pigmentation alone may be responsible for decreased eye size. This difference is also evident in the RNA-seq for *tph* in Figure 4, suggesting that these animals simply have decreased pigment pathways, and this explains smaller eye size. This should be addressed and/or included as a control.
4. Distinguishing discernible from non-discernible eyes. How the authors make this distinction is important to include because it is not described in the methods, yet is used to make the claim that this difference in phenotype is faithfully inherited (Figure 2). Differences between these two morphologies should be quantified at the cellular level because they do seem to have a behavioral difference.
5. Pharynx length (Figure 1) should be assessed based on DAPI-staining rather than in live animals, where it is difficult to determine length in a standardized manner.
6. In Figure 1h, it appears that *G. multidiverticulata* has approximately half the cells of *G. dorotocephala*, but this is not reflected in the quantification in Figure 1i. Were these images scored blindly?
7. In Figure 4c, the quantification of "RNAi score" is vague. What does this mean and how does this metric lend itself to statistical testing? Were biological replicates done? How many animals were tested per replicate experiment? The figure legend does not help; neither do the methods.
8. Throughout the manuscript, no details are provided about how images were collected or how quantification was performed. This is critical because many of the arguments depend on cell density counts and colocalization. For example in Figure 4, is EdU labeling equivalent in both species? Was quantification performed in individual sections? Are these confocal images? How was colocalization determined? Were similarly sized areas quantified? Were biological replicates performed? These are essential details to include.

Minor comments:

- In Figure 2, much of the data is redundant. E, f, and g should be moved to supplement.
- In Figure 6, flip d vertically so that *G. multidiverticulata* is above *G. dorotocephala*.
- In Figure 4, define what the red and blue indicate? In 4b, what is being scored with these numbers (e.g. 8/14) and where is the animal in the right panels? PRN/PC labeling is difficult to understand; integrate with the panels.
- In Figure 6g, how is the body length incorporated into the y axis for opsin and *ppl1*?

Version 1:

Reviewer comments:

Reviewer #2

(Remarks to the Author)

This manuscript has been improved markedly with additional data, revisions, and new analyses. All of my concerns have been sufficiently addressed and I congratulate the authors on their beautiful and fascinating work.

Reviewer #3

(Remarks to the Author)

A major previous criticism was that the authors incorrectly assumed that all FoxA+ cells were progenitors. This is still a concern. The radiation strategy used shows that FoxA cells are not maintained long-term. This finding only indicates that FoxA cells may not live long post-mitotically, but this does not mean that they are progenitors, as the authors state. The language describing these results (lines 346-7) needs to be toned down. Similarly, in the Discussion, lines 465-467 states that "*G. multidiverticulata* produced fate-specified foxA+ neoblasts for the pharynx and *agat-3* + epidermis progenitors in similar numbers to surface planarians". Importantly, the authors never show this data. The language needs to reflect what is

in the paper.

RNAi score. This metric is still poorly explained. If two blind scorers are given the option to score as normal or abnormal, how does this result in a numerical score?? This is still not adequately described in the figure legend or methods.

The references are messed up. Sometimes they include names, some include just numbers.

Which egfr is being referred to on line 492 should be specified.

Other comments have been addressed adequately.

REVIEWER COMMENTS

Reviewer #1 (Remarks to the Author):

The manuscript “Reduced adult stem cell fate specification led to eye reduction in cave planarians” by Saad et al. investigates the mechanisms behind eye loss in the cave planarian species *Girardia multidiverticulata*, focusing on the role of adult stem cells (neoblasts) in this evolutionary adaptation. The study reveals that the reduced eye size in these cave-dwelling planarians is not due to a decrease in overall stem cell numbers but results from a lower rate of stem cell specification to eye progenitors. This finding suggests a novel evolutionary mechanism for trait loss, highlighting the importance of stem cell fate specification in organ size reduction and adaptation to dark environments. I enjoyed reading the paper, it is thoroughly done, well written, and the conclusions are in line with the results. It certainly adds to the knowledge in the field. I have no major concerns, though I would like to know what the author’s thoughts are on why this species shows decreased tissue regeneration despite having apparently no difference in stem cell-derived progenitors for tissues other than the eye.

We thank the reviewer for the interest and positive comments on the work. New recently generated data on mitotic cell numbers in fact confirmed significantly lower mitotic activity during regeneration (after 6h) in the cave-dwelling planarian *Girardia multidiverticulata* compared to the surface species, despite maintaining similar rates of mitosis during homeostasis (Fig. 7j, k). Cave planarians did not exhibit as robust peak of proliferative cells at 6h and 24h associated with regeneration as did surface planarians, but instead mitotic cells remained at steady levels of mitosis (100-400 H3P+/mm²). This data suggests that there was a modification in the proliferative response to injuries in the course of evolution of *G. multidiverticulata*. Possible hypotheses for this include a lower frequency of injury in the cave environment, loss of capacity for asexual reproduction and the associated regenerative proliferative response from a surface ancestor, or some other adaptation to the cave environment. We now discuss this hypothesis in the “Slower regeneration following amputation in cave planarians” results section. Accordingly, we find that cave planarians display slower regeneration after amputation of multiple cell types (Figure 7). However, the ultimate size of tissues ends up similar in *G. multiverticulata* compared to surface species, except for eyes which are smaller, because final tissue size will ultimately be dependent on homeostatic production rate. Eyes also displayed striking delays in initial nucleation in regeneration.

Reviewer #2 (Remarks to the Author):

Eye loss is a common phenomenon in troglobites, but has been largely studied in vertebrates (e.g. fish) and ecdysozoan species (e.g. crustaceans). To see how eye loss has occurred in more distant species (i.e. the spiralian clade), the authors used a species of cave planarians, *Girardia multidiverticulata*, which have reduced eye size. The authors identify two morphotypes of *Girardia multidiverticulata* and find that both maintain small, functional eyespots. The authors also go on to show that much of the genetic program of eye development is shared between *Girardia multidiverticulata* and other planarian species, with an interesting exception of *dlx*. To investigate why *Girardia multidiverticulata* have smaller eyespots, the authors investigated cell birth for eye cells and claim that a lower rate of cell birth explains small eye size.

Strengths of the paper include the identification and broad characterization of eyes in a new species, which required foundation building (e.g. transcriptome creation and adaptation of methods). The work is also thorough, with rigorous quantification and thoughtful data presentation. However, the main claim of slow cell birth specific to eye cells requires more evidence before publication. Specific comments and questions follow.

We thank the reviewer for the feedback and general comment about the need to include more evidence concerning the slow cell birth specific to eye cells. We address his concern carefully by incorporating new evidence as will be described in detail below.

Major comments or questions:

1. The main mechanistic argument made in this paper is that *Girardia multidiverticulata* have smaller eyespots due to a slower rate of cell birth for eye cells. They also present convincing data that fewer eye cells are born in a given period in this species (Fig. 5A/C). However, the EdU+ cell number in general seems much lower in *Girardia multidiverticulata* compared to *Girardia dorotocephala* (Fig. 5B), raising the possibility that the observation is due to an overall low rate of cell birth rather than a specific difference for the eye.

a. Can the authors quantify total number of EdU+ cells/animal at these time points?
Image 5B (Currently 5C) is a single confocal plane showing double positive cells (for *ppl-1* and EdU), and cannot be used to infer the total number of dividing (EdU+) cells in the two species. We have now quantified the number of EdU-positive cells in 100 μm^2 squares (n = 4-6) across different regions of the body in z-stack confocal images and at three time points (4, 8, and 12 days) during homeostasis (Supplementary Fig. 6a, b). The cell counts were comparable between the two species, suggesting that they exhibit similar overall cell incorporation rates during homeostasis.

b. Are the number of stem cells or mitotic stem cells different in the two species (the Smedwi-1 staining in Fig. S5B does look fainter in *Girardia multidiverticulata*).

We analyzed the numbers of H3P+ mitotic cells in intact animals in both species. These data are presented in Fig. 7j, k, and show no significant differences in overall dividing neoblast numbers between cave and surface planarians.

The total number of *smedwi-1*+ cells in a whole animal is very hard to calculate because of ill-defined cytoplasmic staining of the *smedwi-1* signal, and manually counting those cells for the analysis represents a big challenge. To overcome this problem, we calculated the total number of *smedwi-1*+ cells in a specific lateral region of the animal by modeling the expression signal in 3-dimensions using Imaris (Supplementary Fig. 6e, f). From this analysis, we find that the percentages of *smedwi-1* cells are not significantly different between the two species (20-30% for *G. multidiverticulata* and 20-35% for *G. dorocephala*). Altogether these results demonstrate that in homeostatic conditions, cave and surface planarians present a similar number of stem cells and mitotic stem cells.

We also modified the representative image of piwi-1 expression in the unirradiated *G. multidiverticulata* in the previous Fig.S5B for a more representative planarian that attests to a more realistic difference in expression signal before and after irradiation (Supplementary Fig. 6i).

c. Can the authors show that there is no difference in EdU+ cell types in addition to EdU+/ppl-1+ cells (which are present in such low numbers that differences might be obscured)?

To address this concern by the reviewer, we have incorporated new data showing EdU-marker double positive cells of a ciliated neuronal marker *pkd11-2* (EdU+/*pkd11-2*+ cells) (Fig. 5 f, g). A similar incorporation rate was found between cave and surface species for this cell type as well. The overall EdU+ counts, *smedwi-1*+ counts, and H3P counts described above, also further support a similar homeostatic overall cell production level in these species as well. Also see the next answer regarding new data on similar epidermal progenitor numbers for more support for this conclusion.

d. FoxA is present in both progenitors and differentiated cells in Schmidtea and some of the differentiated cells are outside of the pharynx (Adler, 2014). Further, it's possible that FoxA or other progenitor markers might mark cells differently in cave planarians. Can the authors perform double FISH with FoxA and Smedwi-1 (or another progenitor/stem cell marker) to clearly mark pharyngeal progenitors to quantify them? Additionally, the FoxA+ cells counterintuitively look more abundant in *Girardia multidiverticulata* in Fig. 5F, but are shown as not significant in Fig. 5G. Methods for cell counting should be detailed more clearly here and elsewhere.

Unfortunately, we found the double FISH for *FoxA* and *smedwi-1* was insufficient for precisely counting of pharyngeal progenitors.

To circumvent this problem, nonetheless, we performed a new experiment to evaluate whether *FoxA*+ cells could be observed in irradiated animals (lethal

dose, 6 dpi). At this timepoint, most *FoxA* positive cells in the counted trunk region disappeared in both species, suggesting that the vast majority of quantified cells were indeed progenitors. We added this experiment to Supplementary Fig. 6g. We have included a descriptive text of this experiment in the Methods, and changed the *FoxA* representative images for a more realistic representation, as supported by the complete cell counts and statistics.

Furthermore, we complemented these results by comparing the number of *agat-3*-positive cells between the two species. *agat-3* is a marker of late-stage epidermal progenitor cells that is not expressed in mature epidermis, allowing clear distinction between progenitor and mature cell types for a different tissue. Quantification of *agat-3*-positive cells within an equivalent region across animals of similar size revealed that *G. multidiverticulata* and *G. dorotocephala* also possess comparable numbers of epidermal progenitor cells (Fig. 6f, g)

2. In the manuscript, the authors identified two morphotypes that segregate in near-Mendelian ratios and retain their phenotype across amputations, but there is no follow-up on molecular or cellular differences between these genotypes/phenotypes of planarians.

a. Are there genes that are expressed differentially in the *Girardia multidiverticulata* that have visible or non-visible eyes that would explain the difference? I see some data to this effect in Fig. S3 but it might make sense to put it in the main figure if this is a main point of the work.

The only genes that we observed to display significant different levels of expression between the two morphotypes are now shown in main Fig. 4f, and the results are also included in the results section “Eye formation and differentiation programs are largely conserved in *Girardia multidiverticulata*”. There are a small number of differentially expressed genes, including *tyrosinase*, but it is uncertain which particular differences are causal of the phenotype difference.

b. Are there cellular differences, for example in the number of eye cell birth rates in these two morphotypes?

To address this question, we performed the following experiments: In an eye regeneration time course we quantified *opsin+* and *tyrosinase+* cells (Fig. 7a, g-i) and found no differences in eye cell birth rates between the two morphotypes.

We also added information on EdU+ incorporation rates in the eye of both cave morphotypes (Supplementary Fig. 6b), which showed no differences. These experiments demonstrates that eye morphotype differences are not related to eye cell birth rates, but instead rely mainly on pigment synthesis defects, as suggested by the differential gene expression analysis.

c. As a more minor comment, it is then a bit unclear about whether Fig. 4-6 are with one morphotype or with all cave planarians taken together.

We have now clarified this point in the figure legend and text.

3. The authors say that *Girardia multidiverticulata* have slower eye nucleation, but also slower regeneration of the head. Are other head markers (anterior markers/other brain markers) also slower to regenerate? If so, then perhaps the eye nucleation may not be a specific phenotype.

***Girardia multidiverticulata* indeed displayed an overall slower rate of regeneration following amputation (e.g., Fig.7a). A morphological characterization of structures and markers showed a slower rate of regeneration (Fig. 7a-f). New data on mitotic cell numbers revealed significantly lower mitotic activity during regeneration in the cave-dwelling planarian *Girardia multidiverticulata* compared to the surface species (Fig. 7k). Cave planarians did not exhibit as robust of a peak in proliferative cells associated with regeneration as did surface planarians. This data suggests that there was a modification in the proliferative response to injuries in the course of evolution of *G. multidiverticulata*. However, the ultimate size of tissues (brain and pharynx) ends up reaching a similar size in *G. multidiverticulata* compared to surface species at the end of regeneration, except for eyes which are always smaller; this can be explained by final tissue size ultimately being dependent on homeostatic production and turnover rate. Mitotic cell counts eventually equalized between cave and surface planarians under homeostatic conditions.**

4. Does the reduction of photoreceptor neurons in *Girardia multidiverticulata* come with a corresponding decrease in other neurons in the visual circuit within the brain? This might be evident from the RNA-Seq results, focusing on the differentially expressed genes that are not enriched in expression in the eye.

The analysis of anti-Arrestin staining in *Girardia multidiverticulata* reveals a low number of photoreceptor axons forming the optic chiasm and projecting to the brain (Fig. 1d). However, we are not aware of any other neurons at this point that may can definitively be called and assessed as part of the visual circuit. Our RNA-seq analysis was focused on isolated eyes, precluding revealing other differences. Investigating the possibility of other cells being part of the visual circuit and whether there are any changes in these cells in the cave species would be an interesting target for future investigations.

5. Is it possible that *Girardia multidiverticulata* vision is adapted to low light or different wavelengths of light compared to surface species? I wonder if they would perform differently than surface species in a light gradient that is dimmer or if they might be more responsive to long wavelengths of light. I also wonder if *Girardia multidiverticulata* might have retained limited vision to see bioluminescent prey in caves?

We have now included data of a new behavioral experiments that evaluated the responses of cave planarians to red, green, and blue wavelength filters. Our findings revealed that, a lack of negative phototaxis under red light wavelength

treatment in both species. This lack of response was expected as planarians had been shown not to have red wavelength photosensory response (Paskin et al., 2014). But rather unexpectedly we found that in contrast to their surface-dwelling counterparts, *G. multidiverticulata* do not exhibit a pronounced photophobic response to green or blue wavelengths. This lack of a strong color-light avoidance behavior is likely attributed to their diminished eye size and reduced visual capabilities. We have added this data in the text and to figure panel 3d; details on these experiments have also been included in the Methodology. Thus, the cave planarian seems to respond poorly when filters were added, suggesting that they need bright white light to present a photophobic behavior. The question raised by the reviewer related to the possibility that the reminiscent light response in cave planarians may have been preserved to see bioluminescent prey is interesting. However, no bioluminescent species have been observed so far in the cave.

6. Some of the methods were unclear. For differential gene expression experiments, it wasn't clear which figures/experiments used which RNA sequencing strategy. Addition of diagrams in the figures would help the reader sort this out much more easily. It also wasn't clear which figures included data from the single eye purification protocol.

Thank you for this feedback. To address clarity, we added more detail to the methodology in the results and methods sections. Furthermore, we included an RNA sequencing schematic pipeline to enhance comprehension of the experiment (Supplementary Fig. 4d).

Minor comments:

1. *Dugesia* is misspelled in some of the figures.

We appreciate you bringing this to our attention. The genus name has been corrected

2. Occasionally, sample labels are missing from figures (e.g. Fig. 2e, f, g).

Sample labels were corrected

3. I appreciate the data presentation in Fig. 2 but the black averages are not always visible in this format. Could these be modified slightly to be clearer?

We now brought the averages to the front of the figure to improve visualization

4. Given that there are multiple SoxB genes in the Schmidtea genome, it would be helpful to identify which one this is. Is the SoxB here similar to SoxB in Lapan which was renamed SoxB1-1 in Ross (2018)?

We now corrected the gene name in the text, which corresponds to SoxB1-1

5. While I appreciate the complexity of the data shown in Supp. Fig. S3, it is hard as a reader to make meaning of the data with all individual datapoints shown. Averages for

each gene with notation of statistic differences might be easier to parse. Likewise, Supp. Fig. S4 lacks notation of whether statistical differences were present between any sample, so it is challenging to draw inferences from the data.

We have now included heatmaps in the main figure (Fig. 4e), focusing on genes with an adjusted p-value less than 0.05 and an absolute log₂ fold change greater than or equal to 1. Additionally, we have complemented Supplementary Table 3 with all data from the pair-wise DESeq2 Wald test comparisons for clearer results.

6. Full sequences for all *Girardia multidiverticulata* genes used in this study should be included in a supplemental figure and deposited so that others could replicate

We have now added the full sequence for all *Girardia multidiverticulata* eye genes used in this study. Supplementary Data 1. Gene sequences will be also be available through Genbank.

Reviewer #3 (Remarks to the Author):

This manuscript from Saad et al describes a potential new mechanism underlying the small-sized eyes observed in cave-dwelling planarians from Brazil. Borrowing from the molecular and cellular description of eye composition in *S. mediterranea*, the authors characterize the cellular composition of *G. multidiverticulata* eyes, demonstrate that despite their small size, they retain residual function in light aversion, perform a comparative transcriptome analysis of cave versus surface planarian eyes, and describe eye regeneration. Some experiments are quite convincing. For example, the behavioral characterization shows that these rudimentary eyes are still sufficient to mediate light avoidance is clear and includes convincing controls. Other experiments and their conclusions have issues (comments below).

We would like to thank the reviewer for his feedback and comments for improvement. We have addressed all his comments as described below point-by-point.

Ultimately the model provided is that stem cells in *G. multidiverticulata* have a decreased probability to adopt a photoreceptor fate. This model leans heavily on what's known in *S. mediterranea*, where extensive characterization of stem cells and their descendants has shown that the plentiful stem cells can adopt a variety of organ specific cell fates, although it remains unclear whether this happens at the level of a single cell or collectively among multiple stem cells. This distinction is muddled in the writing. However, another possibility is that the stem cell population as a whole is programmed to churn out organ specific descendants at a certain rate. This rate seems to be overall slowed down in *G. multidiverticulata* as compared to surface planarians. This is supported by data in Figure 6, and should be addressed.

To address this point, we have made additional experiments, including the generation of new data and analyses on EdU, H3P, neoblast numbers, and progenitor numbers of different tissues and organs. For specific details, please read our reply to comment 1 below.

Another issue with this model is that the authors state that neoblasts 'make the choice' to become an eye progenitor at a lower rate than in surface planarians. This conclusion partly depends on the presence of FoxA cells being equivalent in *G. multidiverticulata* and *G. dorotocephala* (model figure). However, the data used to support this, in particular with respect to FoxA, is incorrect. In *S. mediterranea* it's known that not all FoxA-positive cells are stem cells. To make this claim about FoxA cells being stem cells, either EdU labeling or co-labeling FoxA along with a stem cell marker would be necessary. Otherwise, it should be removed.

We performed new experiments that continue to support the conclusion that the vast majority of *foxA+* cells correspond to pharynx-fated progenitors, and that these acts similarly in both *G. multidiverticulata* and *G. dorotocephala*. We irradiated animals at neoblast depletion dosage, and at 6 days post irradiation

counted for *foxA+* cells. In both species, most *foxA+* cells had disappeared from the trunk region, suggesting that these cells are predominantly progenitor cells. We have incorporated the results of this experiment in the text and into Supplementary Fig. 6g.

Additionally, we sought to complement this experiment by comparing the depletion of another progenitor population by irradiation and using a different marker, which could be clearly distinguished from the terminally differentiated cells. We used *agat-3* as a marker of late-stage epidermal progenitor cells, which is not expressed in mature epidermis, and compared the number of *agat-3*-positive cells between *G. multidiverticulata* and *G. dorotocephala*. Quantification of *agat-3*-positive cells in equivalent subepidermal regions across animals of similar size revealed that the two species possess comparable numbers of epidermal progenitor cells (Fig. 6f-g).

Below are major and minor comments that will help to clarify the data presentation and provide more evidence regarding stem cell behaviors being responsible for the morphological difference in *G. multidiverticulata*.

1. The data suggests that regeneration in this species occurs more slowly than surface-dwelling planarians, suggesting an alternative possibility which is that stem cells are in general just slower to respond both to injury and death. Basic characterization of stem cell dynamics in the two species should be included, such as basal cell proliferation or differentiation rates (with phosphohistone H3 and/or EdU) and injury-induced proliferation responses.

We have now included all requested data on the proliferation dynamics, differentiation rates, and injury induced proliferation responses in the cave and surface planarians. Our results revealed a similar baseline level of proliferation (H3P+ cells) between *Girardia multidiverticulata* and surface species (Fig. 6j-k). We also quantified the overall levels of EdU incorporation into tissues generically by counting the number of EdU-positive cells within a 100 x 100 μm square region of the animals at different time points (Supplementary Fig. 6a, b). The cell counts were comparable between the two species, suggesting that they exhibit similar overall cell incorporation rates during homeostasis. We also included new data on the overall number of neoblasts between the two species. We measured the total number of *smedwi-1+* cells anterior to the pharynx and laterally between the pharynx and the edge of the animal by modeling images in 3-dimensions using Imaris. The percentage of *smedwi-1+* cells in this region was similar between the two species (Supplementary Fig. 6e, f). Furthermore, there were a similar number of *agat-3* epidermal progenitors in the cave and surface species (Fig. 6f, g), also consistent with a similar overall cell production rate. Finally, we also showed similar EdU incorporation rates into a second brain cell type (*pkd11-2+* neurons, Fig. 5f, g).

With regards to amputation responses, we found significantly lower mitotic activity during regeneration in the cave-dwelling planarian *Girardia*

multidiverticulata compared to the surface species (Fig. 7j, k). The reduced number of mitotic cells during the second peak of regeneration appears to account for the slower rate of formation of brain cells, pharynx, and eyes observed in the data. Eventually, homeostatic proliferation rates between the two species normalize. The ultimate size of tissues (brain and pharynx) ends up reaching a similar size in *G. multidiverticulata* compared to surface species at the end of regeneration; the eyes however, always remain smaller. This can be explained by final organ size ultimately being dependent on homeostatic production and turnover rate. This phenomenon was demonstrated in *folliculin* RNAi animals in *S. mediterranea*: animals without a missing tissue proliferative response during regeneration underwent the process slower, but ultimately reached normal organ size likely based on homeostatic processes of cell proliferation. The reason for a lower proliferative response to injury and lack of acceleration of regeneration in *G. multidiverticulata* is interesting but the molecular and cellular mechanisms that regulate it remain unknown. We have now commented this in the manuscript text.

2. Much of the argument that *G. multidiverticulata* and *G. dorotocephala* differ in their stem cell-specific differentiation rests on evidence that there are key changes occurring at the transcriptional level. The RNA sequencing in Figure 4d shows that on average, there may be a difference, but it appears that the violin plots are strongly skewed by a couple of outlier data points (in ovo, *six1/2*, *dlx*, *foxQ2*). Are these outlier points from the same samples? It is important to address this because on average, *ovo* expression in *G. multidiverticulata* and *G. dorotocephala* is overall quite similar except for these outliers; it's discordant with the in-situ expression in Figure 5e.

We have included a text of clarification of our data in the revised results section entitled "Eye formation and differentiation programs are largely conserved in *Girardia multidiverticulata*".

For clarification, the violin plots display the average expression levels of eye-related genes across the different species, but do not represent statistical significance. Notably, the violin plot distribution of average expression for *ovo*, *six1/2*, *dlx*, and *foxQ2* is indeed similar between cave and surface species. Complementarily, the differential gene expression results corroborate these findings, confirming the lack of significant p-value differences in eye-related transcriptional factors between cave and surface planarians ($p_{adj} < 0.05$ and $\log_2\text{FoldChange} > 1$, $\log_2\text{FoldChange} < -1$) (Supplementary Table 3). In the current revision, we have included the heatmaps to highlight the genes that presented statistical differences as a main figure (Fig. 4e).

Our RNA sequencing data shows that *ovo* expression in the eye cells, present after the count normalization performed by DESeq2, is comparable between the two species despite the lower number of eye cells in the cave species. This apparent disagreement in the data can be explained because RNA extractions for sequencing derived from isolated eyes that did not include more distantly located

progenitors. In Figure 6b, we can appreciate a number of ovo-positive cells located outside of the eye in cave planarians that were not sampled for sequencing. These observations suggest that whereas cave planarians exhibit similar expression of eye transcription factors—within differentiated eye cells themselves—to surface species, the eye stem cell fate specification process presents differences.

3. *G. multidiverticulata* animals appear to be overall lighter in color than other surface-dwelling planarians, suggesting that differences in pigmentation alone may be responsible for decreased eye size. This difference is also evident in the RNA-seq for *tph* in Figure 4, suggesting that these animals simply have decreased pigment pathways, and this explains smaller eye size. This should be addressed and/or included as a control.

We agree with the reviewer that the cave planarian *Girardia multidiverticulata* has not only lost body pigmentation in both morphotypes, but may have lost additional eye-specific pigmentation in the non-discernible eye morphotype. Based on published data from the related species *Schmidtea mediterranea*, the progenitor cells for eye pigment cup cells and body pigments are distinct^{1,2}. Whereas eye optic pigment is melanin, body pigment is a different pigment³⁻⁵. We believe that the decrease of *tph* observed in the differential gene expression could be associated with the lack of pigmentation in the optic cup, but have no reason to believe that it could account for the overall smaller eye size or reduced number of photoreceptor cells; nor do we have a reason to believe that the loss of body pigmentation could be directly influencing eye size in the cave planarian. For example, the *tph* gene knockdown affects eye pigment synthesis, without affecting optic cup formation, and also does not impact body pigmentation³. The dynamics of body and eye pigment cup cell pigmentation in cave planarians continues to be an interesting topic that could be addressed in future studies.

4. Distinguishing discernible from non-discernible eyes. How the authors make this distinction is important to include because it is not described in the methods, yet is used to make the claim that this difference in phenotype is faithfully inherited (Figure 2). Differences between these two morphologies should be quantified at the cellular level because they do seem to have a behavioral difference.

We have now added a section on the Results and also in the Methods section “Quantification and Statistical Analysis” to explain how we distinguish both morphotypes: “...However, observations of the offspring from animals raised in the laboratory revealed the occurrence of two visually distinguishable morphotypes segregating among siblings. The ‘discernible eye morphotype’ displayed small, pigmented rudimentary eyes visible with light microscopy, whereas the ‘non-discernible eye morphotype’ exhibited no visible eyes by light microscopy”.

We have quantified both cellular and behavioral differences in the two morphotypes (Fig. 2 and Fig. 3a and 3c). The two cave morphotypes display no differences in the number of photoreceptor cells (Fig. 2d). Also, the behavioral assay revealed mostly not significant differences in their behavioral responses to light, with both morphotypes displaying mostly negative phototaxis (Fig. 3a). The only behavioral differences observed between morphotypes was in the eye-resected assays, in which phototaxis and a preference for darker regions was observed in *G. multidiverticulata* individuals with non-discernible eyes. We attribute this photosensitivity to partial or incomplete resections of the eyes as it is technically very challenging to completely remove eyes that are not visible. Thus, resections of the not discernible eyes were guided only by their relative positions in the head, resulting in the residual light sensitivity observed (Fig. 3b), and also likely in the color sensitive responses in 3c. This possibility has been mentioned and discussed in the Results section of the revised manuscript.

Furthermore, we have now included a pair-wise differential gene expression analysis from the RNA sequencing of isolated eyes of both morphotypes to address gene expression differences. Results are now in the main Fig. 4f and discussed in the Results section.

5. Pharynx length (Figure 1) should be assessed based on DAPI-staining rather than in live animals, where it is difficult to determine length in a standardized manner.

We chose to use live images because fixed animals sometimes acquire posture differences during the fixation process affecting size measurements. However, we now also provide information on pharynx length based on DAPI-staining (Supplementary Fig. 2a-c), which demonstrates similar results to those obtained from live pharynx images, indicating no significant differences in pharynx length between surface and cave species.

6. In Figure 1h, it appears that *G. multidiverticulata* has approximately half the cells of *G. dorotocephala*, but this is not reflected in the quantification in Figure 1i. Were these images scored blindly?

The images were indeed blindly scored, we added more information about data acquisition on the Methods “Quantification and Statistical Analysis” section. We changed the *ppl1*⁺ cells image for a more representative picture. Despite the variation on *ppl1*⁺ cells numbers among the ~16 specimens analyzed, the overall quantification showed no differences between the cave and surface species (Fig. 2f).

7. In Figure 4c, the quantification of “RNAi score” is vague. What does this mean and how does this metric lend itself to statistical testing? Were biological replicates done? How many animals were tested per replicate experiment? The figure legend does not help; neither do the methods.

We thank the reviewer for noticing this. Indeed, the definition was vague, and we have now improved the text and included all the requested information by the reviewer in the results, figure legends, and methods with clearer explanations of the analysis. For example, the text in the Results now reads: “RNA interference (RNAi) experiments showed that *ovo*, *six-1/2-1*, and *eya* are necessary for eye formation in *G. multidiverticulata*, because down-regulation of these genes resulted in animals without eyes or with malformed eyes (Fig. 4b). More specifically, *eya* inhibition resulted in complete absence of eyes in all treated animals (n=10). Inhibition of the other central regulators of eye formation (*ovo*, *six-1/2-1*) resulted in a slightly different outcome. Some RNAi animals exhibited a complete absence of eyes (Fig. 4b, *ovo* and *six-1/2-1* left panels) or the presence of a small number of photoreceptor cells (~one or two Arrestin⁺ cells), with severe anatomical malformation (Fig. 4b, *ovo* and *six-1/2-1* right panels). By contrast, inhibition of the genes encoding other eye-associated transcription factor-encoding genes, *dlx*, *otxA*, and *foxQ2*, did not block eye formation entirely, but instead resulted in eyes that were frequently malformed (Fig 4c). Unexpectedly, RNAi of the *dlx* gene affected not only the development of photoreceptor cells, but also the optic cup cells (Fig. 4c). Additionally, RNAi of the *foxQ2* gene, which was previously reported to impact only the number of photoreceptor cells (Lapan & Reddien, 2012), was found to influence overall eye formation in cave-dwelling planarians, resulting in abnormal eyes presenting few photoreceptor cells, asymmetric eyes, or misshapen eyes. Eyes from control, *six1/2-1*, *dlx*, *ovo*, *otxA*, and *foxQ2* RNAi animals were classified as either normal or abnormal by two independent scorers, who were blinded to condition and results were compared to random predictions with a Fisher's exact test (Fig. 4c).”

8. Throughout the manuscript, no details are provided about how images were collected or how quantification was performed. This is critical because many of the arguments depend on cell density counts and colocalization. For example in Figure 4, is EdU labeling equivalent in both species? Was quantification performed in individual sections? Are these confocal images? How was colocalization determined? Were similarly sized areas quantified? Were biological replicates performed? These are essential details to include.

We have revised all sections and added detailed information for each experiment performed congruent with the reviewer's requests, either in the main text, the legends, or in the Methods “Quantification and Statistical Analysis” section.

Minor comments:

– In Figure 2, much of the data is redundant. E, f, and g should be moved to supplement.

Figures can now be found in Supplementary Fig.2

– In Figure 6, flip d vertically so that *G. multidiverticulata* is above *G. dorotocephala*.

Figures were flipped as suggested

– In Figure 4, define what the red and blue indicate? In 4b, what is being scored with these numbers (e.g. 8/14) and where is the animal in the right panels? PRN/PC labeling is difficult to understand; integrate with the panels.

We have added a better explanation of the methods used in the results, figure legend, and methods. For example, we have made the following amendments to the Fig. 4 legend: “...b) RNAi experiment phenotypes presenting normal (control), abnormal (*ovo*, *six1/2* right panels), or absent (*eya*, *ovo*, *six1/2* left panels) eyes. In control RNAi animals, 26 out of 26 presented normal eyes (26/26). Normal eyes contained photoreceptor neurons and photoreceptor axons visualized with the anti-Arrestin (VC-1) antibody. Six out of 14 *ovo* RNAi animals displayed no eye formation (6/14), and eight out of 14 displayed an abnormal eye (8/14), containing few photoreceptor cells and disorganized axon projections. Similarly, some treated animals with *six1/2* RNAi displayed no eye formation (6/17), but the majority displayed abnormal eyes (11/17). c) Abnormal phenotypes after eye transcription factor knockdown by RNAi. Eye images shown of each experimental condition: Control ($n = 26$), *six1/2* ($n = 17$), *dlx* ($n = 8$), *ovo* ($n = 22$), *otxA* ($n = 16$), and *foxQ2* ($n = 11$). RNAi during regeneration and homeostasis were blindly scored by two independent scorers, who classified the images as either normal (green) or abnormal (blue). The resulting scores were then compared using a Fisher's exact test, ** $p < 0.01$, * $p < 0.001$, **** $p < 0.0001$. RNAi experiments were performed with the *G. multidiverticulata* non-discernable eye morphotype.”**

– In Figure 6g, how is the body length incorporated into the y axis for opsin and *ppl1*?

In this analysis, we used animals of comparable size (matching size). Accordingly, we did not incorporate body length as a variable in the analysis. We have corrected and clarified this point in the figure axis, text and legend.

References

1. Stubenhaus, B. M. *et al.* Light-induced depigmentation in planarians models the pathophysiology of acute porphyrias. *Elife* **5**, (2016).
2. Lindsay-Mosher, N. & Pearson, B. J. The true colours of the flatworm: Mechanisms of pigment biosynthesis and pigment cell lineage development in planarians. *Semin Cell Dev Biol* **87**, 37–44 (2019).
3. Lambrus, B. G. *et al.* Tryptophan hydroxylase is required for eye melanogenesis in the planarian *Schmidtea mediterranea*. *PLoS One* **10**, (2015).
4. Lapan, S. W. & Reddien, P. W. Transcriptome analysis of the planarian eye identifies *ovo* as a specific regulator of eye regeneration. *Cell Rep* **2**, 294–307 (2012).
5. Lapan, S. W. & Reddien, P. W. *dlx* and *sp6-9* control optic cup regeneration in a prototypic eye. *PLoS Genet* **7**, e1002226 (2011).

REVIEWERS' COMMENTS

Reviewer #2 (Remarks to the Author):

This manuscript has been improved markedly with additional data, revisions, and new analyses. All of my concerns have been sufficiently addressed and I congratulate the authors on their beautiful and fascinating work.

Reviewer #3 (Remarks to the Author):

A major previous criticism was that the authors incorrectly assumed that all FoxA+ cells were progenitors. This is still a concern. The radiation strategy used shows that FoxA cells are not maintained long-term. This finding only indicates that FoxA cells may not live long post-mitotically, but this does not mean that they are progenitors, as the authors state. The language describing these results (lines 346-7) needs to be toned down. Similarly, in the Discussion, lines 465-467 states that “*G. multidiverticulata* produced fate-specified foxA+ neoblasts for the pharynx and agat-3 + epidermis progenitors in similar numbers to surface planarians”. Importantly, the authors never show this data. The language needs to reflect what is in the paper.

We modified the text further on this point.

Results: “ To determine whether the decreased number of eye progenitors was specific to the eye stem cell fate, or could alternatively reflect a lower overall rate of progenitor production, we assessed pharynx progenitor production using a probe to *FoxA*. *FoxA* is expressed in a subset of pharyngeal neoblasts and also in some differentiated cells, prominently in the pharynx itself^{74–76}(Fig. 7d, Supplementary Fig. 7a). The number of *FoxA*⁺ presumptive pharynx progenitor cells counted in a region just anterior to the pharynx but excluding the pharynx itself was similar between *G. multidiverticulata* and *G. dorotocephala* (Fig. 7d,e and Supplementary Fig. 7a). *FoxA*⁺ cells in this region are known to prominently include pharynx progenitors, but other *FoxA*⁺ cells could possibly be present. Neoblasts can be depleted largely specifically by irradiation (Supplementary Fig. 7b)⁷⁷ and *G. multidiverticulata* animals four days post-irradiation exhibited a strong reduction in *FoxA*⁺ cells in this region, consistent with an interpretation that counted cells prominently included presumptive pharynx progenitors (Supplementary Fig. 7a).”

Discussion: “Fewer eye-specialized neoblasts were present in uninjured *G. multidiverticulata* when compared to surface species. By contrast, *G. multidiverticulata* produced presumptive fate-specified foxA+ neoblasts for the pharynx and agat-3+ epidermis progenitors in similar numbers to surface planarians.”

Note that no differentiated cells have been described that are this acutely irradiation sensitive (i.e., the irradiation sensitivity of expression and prior work on *FoxA* strongly support the view that the majority of cells examined are

pharynx progenitors).

RNAi score. This metric is still poorly explained. If two blind scorers are given the option to score as normal or abnormal, how does this result in a numerical score?? This is still not adequately described in the figure legend or methods.

We believed that the term RNAi score may have cause some confusion. We modified the text and graphs to read the "number of animals". We also added a better explanation to the methods: "Each picture analyzed was classified as either normal or abnormal by two blinded examiners. These results were then compared to random predictions using a Fisher's exact test (p<0.01)"

The references are messed up. Sometimes they include names, some include just numbers.

Thank you for pointing this out, the references are now fixed

Which *egfr* is being referred to on line 492 should be specified.

We now made it clear which *egfr* gene is being referred to: "Inhibition of *egfr-4* increases the number of eye progenitor cells at the expense of differentiated eye cells, resulting in smaller eyes"

Other comments have been addressed adequately.